# DCC regulates astroglial development essential for telencephalic morphogenesis and corpus callosum formation

Laura Morcom[1†], Ilan Gobius[1‡], Ashley PL Marsh[2,3§], Rodrigo Suárez[1#], Jonathan WC Lim[1], Caitlin Bridges[1], Yunan Ye[1], Laura R Fenlon[1#], Yvrick Zagar[4], Amelia M Douglass[1¶], Amber-Lee S Donahoo[1], Thomas Fothergill[1**], Samreen Shaikh[1#], Peter Kozulin[1], Timothy J Edwards[1,5††], Helen M Cooper[1], IRC5 Consortium[6], Elliott H Sherr[7], Alain Chédotal[4], Richard J Leventer[3,8,9], Paul J Lockhart[2,3], Linda J Richards[1,10]*

[1]The University of Queensland, Queensland Brain Institute, Brisbane, Australia; [2]Bruce Lefroy Centre for Genetic Health Research, Murdoch Children's Research Institute, Royal Children's Hospital, Parkville, Australia; [3]Department of Paediatrics, University of Melbourne, Parkville, Australia; [4]Sorbonne Université, INSERM, CNRS, Institut de la Vision, Paris, France; [5]The University of Queensland, Faculty of Medicine, Brisbane, Australia; [6]Members and Affiliates of the International Research Consortium for the Corpus Callosum and Cerebral Connectivity (IRC5), Los Angeles, United States; [7]Departments of Neurology and Pediatrics, Institute of Human Genetics and Weill Institute of Neurosciences, University of California, San Francisco, San Francisco, United States; [8]Neuroscience Research Group, Murdoch Children's Research Institute, Parkville, Australia; [9]Department of Neurology, University of Melbourne, Royal Children's Hospital, Parkville, Australia; [10]The University of Queensland, School of Biomedical Sciences, Brisbane, Australia

*For correspondence:
richards@uq.edu.au

Present address: [†]The University of Cambridge, Department of Paediatrics, Wellcome-MRC Stem Cell Institute, Cambridge, United Kingdom; [‡]The University of Queensland, Diamantina Institute, Brisbane, Australia; [§]Departments of Child Health, Neurology, Cellular and Molecular Medicine and Program in Genetics, University of Arizona College of Medicine, Phoenix, United States; [#]The University of Queensland, School of Biomedical Sciences, Brisbane, Australia; [¶]Division of Endocrinology, Diabetes, and Metabolism, Department of Medicine, Beth Israel Deaconess Medical Center, Harvard Medical School, Boston, United States; [**]Bruker Fluorescence Microscopy, Madison, United States; [††]Great Ormond Street Institute of Child Health, University College London, London, United Kingdom

Competing interests: The authors declare that no competing interests exist.

**Abstract** The forebrain hemispheres are predominantly separated during embryogenesis by the interhemispheric fissure (IHF). Radial astroglia remodel the IHF to form a continuous substrate between the hemispheres for midline crossing of the corpus callosum (CC) and hippocampal commissure (HC). Deleted in colorectal carcinoma (DCC) and netrin 1 (NTN1) are molecules that have an evolutionarily conserved function in commissural axon guidance. The CC and HC are absent in *Dcc* and *Ntn1* knockout mice, while other commissures are only partially affected, suggesting an additional aetiology in forebrain commissure formation. Here, we find that these molecules play a critical role in regulating astroglial development and IHF remodelling during CC and HC formation. Human subjects with *DCC* mutations display disrupted IHF remodelling associated with CC and HC malformations. Thus, axon guidance molecules such as DCC and NTN1 first regulate the formation of a midline substrate for dorsal commissures prior to their role in regulating axonal growth and guidance across it.

## Introduction

The corpus callosum (CC) is the largest fibre tract in the human brain and comprises approximately 200 million axons (*Paul et al., 2007*; *Tomasch, 1954*) connecting similar regions between the left and right cerebral hemispheres (*Fenlon and Richards, 2015*; *Fenlon et al., 2017*; *Suárez et al., 2018*). All eutherian mammals have a CC (*Suárez et al., 2014a*; *Suarez, 2017*; *Suárez et al., 2018*), with malformations or complete absence (agenesis) of the CC occurring in at least 1 in 4000 human

live births (*Glass et al., 2008*). Collectively, these genetically heterogeneous disorders are known as CC dysgenesis and can result in a wide spectrum of neurological, developmental and cognitive deficits (*Brown and Paul, 2019*; *Edwards et al., 2014*; *Paul et al., 2007*).

During brain development, the callosal tract forms between the two telencephalic hemispheres through a midline region initially separated by the interhemispheric fissure (IHF; *Gobius et al., 2016*; *Rakic and Yakovlev, 1968*; *Silver et al., 1982*). Recently, we demonstrated that remodelling of the IHF tissue by specialized astroglial cells, known as midline zipper glia (MZG; *Silver et al., 1993*), is mediated by FGF8 signalling and subsequent regulation of astrogliogenesis by nuclear factor I (NFI) transcription factors and is essential to provide a permissive substrate for callosal axons to cross the telencephalic midline (*Gobius et al., 2016*). MZG are derived from radial glia in the telencephalic hinge, located rostral to the third ventricle. From this ventricular zone, they migrate rostro-dorsally as bipolar cells to the IHF pial surface and transition into multipolar astrocytes. This latter step facilitates their intercalation across the midline and subsequent elimination of the intervening leptomeningeal tissue that comprises the IHF. The MZG thereby fuse the medial septum in a fashion that resembles a 'zipper' mechanism (*Gobius et al., 2016*), which does not occur in naturally acallosal mammals such as monotremes and marsupials (*Gobius et al., 2016*). Developmental defects in IHF remodelling invariably result in callosal agenesis in mouse models, and, strikingly, all 38 individuals in a human cohort with callosal agenesis also displayed aberrant retention of the IHF and an abnormal separation of the medial septum (*Gobius et al., 2016*). Thus, the remarkably high prevalence of midline defects in human callosal disorders suggests that there are additional determinant genes for IHF remodelling that have not yet been identified. These could include axon guidance genes, which are frequently mutated in humans (and mice) with CC abnormalities (*Edwards et al., 2014*).

Netrin 1 (NTN1) is a secreted ligand for the deleted in colorectal carcinoma (DCC) receptor, and these molecules function as axon guidance cues in species ranging from *Drosophila* to mammals (*Chan et al., 1996*; *de la Torre et al., 1997*; *Fazeli et al., 1997*; *Hedgecock et al., 1990*; *Keino-Masu et al., 1996*; *Kolodziej et al., 1996*; *Serafini et al., 1996*). Indeed, NTN1-DCC signalling attracts pioneering callosal axons towards the midline and attenuates chemorepulsive signalling in neocortical callosal axons ex vivo to facilitate crossing the midline (*Fothergill et al., 2014*). Heterozygous and homozygous *DCC* pathogenic variants also result in human callosal dysgenesis at high frequency (*Jamuar et al., 2017*; *Marsh et al., 2018*; *Marsh et al., 2017*) with an estimated incidence of 1 in 14 in unrelated individuals with callosal dysgenesis (*Marsh et al., 2017*), and *Ntn1* and *Dcc* mouse mutants do not form a CC (*Fazeli et al., 1997*; *Finger et al., 2002*; *Fothergill et al., 2014*; *Serafini et al., 1996*). Instead of crossing the midline, callosal axons in *Ntn1* and *Dcc* mutant mice form ipsilateral 'Probst' bundles that run parallel to the midline (*Fazeli et al., 1997*; *Finger et al., 2002*; *Fothergill et al., 2014*; *Probst, 1901*; *Ren et al., 2007*; *Serafini et al., 1996*). Together, these results have led to the conclusion that NTN1 and DCC act primarily as axon guidance genes during callosal formation. However, in *Ntn1* and *Dcc* mutant mice, only the CC and hippocampal commissure (HC) are completely absent, while other axon tracts remain intact or are mildly affected (*Fazeli et al., 1997*; *Serafini et al., 1996*; *Yung et al., 2015*), indicating that additional processes might affect the development of the CC and HC in these mice. Moreover, elimination of the leptomeninges, which normally occurs during IHF remodelling (*Gobius et al., 2016*), is severely disrupted in *Ntn1* mutant mice (*Hakanen and Salminen, 2015*), further suggesting that NTN1 and its receptor, DCC, may play a hitherto unidentified role in IHF tissue remodelling.

Here, we identify a distinct and developmentally earlier role for NTN1 and DCC signalling during CC formation, involving the regulation of MZG development and subsequent IHF remodelling. We find that IHF remodelling is impaired in both *Ntn1* and *Dcc* mouse mutants, as well as in humans with *DCC* pathogenic variants that also display agenesis of the CC and HC. Moreover, in contrast to the wildtype receptor, these human pathogenic variants of *DCC* are unable to regulate cell morphology. Furthermore, we find that defects in astroglial morphology and migration to the IHF in *Ntn1* and *Dcc* mutant mice prevent MZG intercalation and, therefore, IHF remodelling and midline crossing of commissural axons. Taken together, our findings indicate that pathogenic variants in *NTN1* and *DCC* are most likely to affect human CC and HC development through misregulation of astroglial shape, motility and function during IHF remodelling.

## Results

### DCC signalling is required for IHF remodelling and subsequent CC and HC formation

To re-investigate how *Dcc* and *Ntn1* regulate callosal formation, we first analysed the relationship between the IHF and callosal axon growth during midline development in horizontal sections of *Ntn1* and *Dcc* mutant mice. These mouse mutants include *Dcc* knockout, *Dcc^kanga* mice, which express a truncated DCC receptor that lacks the P3 intracellular signalling domain, and *Ntn1-lacZ* mutant mice, which express reduced levels of NTN1 protein that subsequently becomes sequestered in intracellular organelles (*Fazeli et al., 1997*; *Finger et al., 2002*; *Fothergill et al., 2014*; *Serafini et al., 1996*). Immunohistochemistry was performed following commissure formation at embryonic day (E)17 against the axonal marker Gap43 together with pan-Laminin, which labels both leptomeningeal fibroblasts and the basement membrane surrounding the IHF (*Figure 1A*). This revealed that commissural axons in *Dcc* knockout, *Dcc^kanga*, and *Ntn1-lacZ* mice remain within the ipsilateral hemisphere and do not form a CC or HC, consistent with previous reports (*Fazeli et al., 1997*; *Finger et al., 2002*; *Fothergill et al., 2014*; *Ren et al., 2007*; *Serafini et al., 1996*). We further identified that IHF remodelling had not occurred in *Dcc* knockout, *Dcc^kanga*, and *Ntn1-lacZ* mice, evidenced by complete retention of the IHF, which separated the majority of the telencephalic midline (*Figure 1A*). This likely prevented formation of the HC in addition to the CC (*Figure 1A*). The extent of IHF retention, measured as the ratio of IHF length to total midline length, is significantly larger in *Dcc* and *Ntn1* mutants compared to their wildtype littermates (*Supplementary file 1*; *Figure 1A, B*), but did not differ between mutants (*Supplementary file 1*; *Figure 1A, B*). This suggests that NTN1 and DCC may interact or act in a similar manner to regulate IHF remodelling prior to commissural axon crossing, and that the P3 intracellular domain of DCC is crucial for this function. The brain phenotype of adult *Dcc* knockout and *Ntn1-lacZ* mice was unable to be investigated as these animals die shortly after birth (*Fazeli et al., 1997*; *Finger et al., 2002*; *Serafini et al., 1996*). However, immunohistochemistry for the axonal marker neurofilament in adult *Dcc^kanga* mice revealed that the retention of the IHF and absence of the CC and HC persists into adulthood (*Supplementary file 1*; *Figure 1B*, *Figure 1—figure supplement 1*), resembling human congenital callosal agenesis (*Edwards et al., 2014*; *Gobius et al., 2016*).

We previously reported that humans carrying *DCC* pathogenic variants develop dysgenesis of the CC with incomplete penetrance (*Marsh et al., 2017*). T1-weighted MRI of four individuals from two unrelated families carrying missense pathogenic variants in *DCC* (p.Val793Gly affecting fibronectin type III-like domain 4 of *DCC* and p.Met1217Val; p.Ala1250Thr affecting the cytoplasmic domain of *DCC*; Figure 8A; *Marsh et al., 2017*) revealed in all cases that the complete absence of the CC was associated with aberrant retention of the IHF and an unfused septum (*Figure 1C*). Importantly, these individuals were also previously reported to lack a HC (*Marsh et al., 2017*), suggesting that a defect in IHF remodelling may also impact HC development. Since IHF remodelling is required for subsequent callosal axon crossing (*Gobius et al., 2016*), these results collectively suggest that the underlying cause of callosal agenesis in *Ntn1* and *Dcc* mutant mice and in humans with *DCC* mutations is a failure of IHF remodelling.

### DCC and NTN1 are expressed by MZG cells throughout interhemispheric remodelling

We previously demonstrated that DCC is expressed on axons of the CC, HC and the fornix during midline development, while NTN1 is expressed at the telencephalic midline, within the indusium griseum and the septum but not within callosal axons themselves (*Fothergill et al., 2014*; *Shu et al., 2000*). Since our analysis of *Ntn1* and *Dcc* mutant mice revealed that these genes are necessary for IHF remodelling, we then investigated whether they are expressed by the MZG, which mediate IHF remodelling (*Gobius et al., 2016*). MZG arise in the telencephalic hinge, a region in the septal midline caudal to the IHF and rostral to the third ventricle. Radial glia within the telencephalic hinge are attached to both the third ventricle and the IHF and mature into MZG as they undergo somal translocation to the IHF between E12 and E16 in mice (*Gobius et al., 2016*). Fluorescent in situ hybridization for *Dcc* and *Ntn1* transcripts, combined with immunohistochemistry for the MZG marker Glast (*Gobius et al., 2016*), revealed *Dcc* and *Ntn1* expression in radial MZG progenitor cells within the

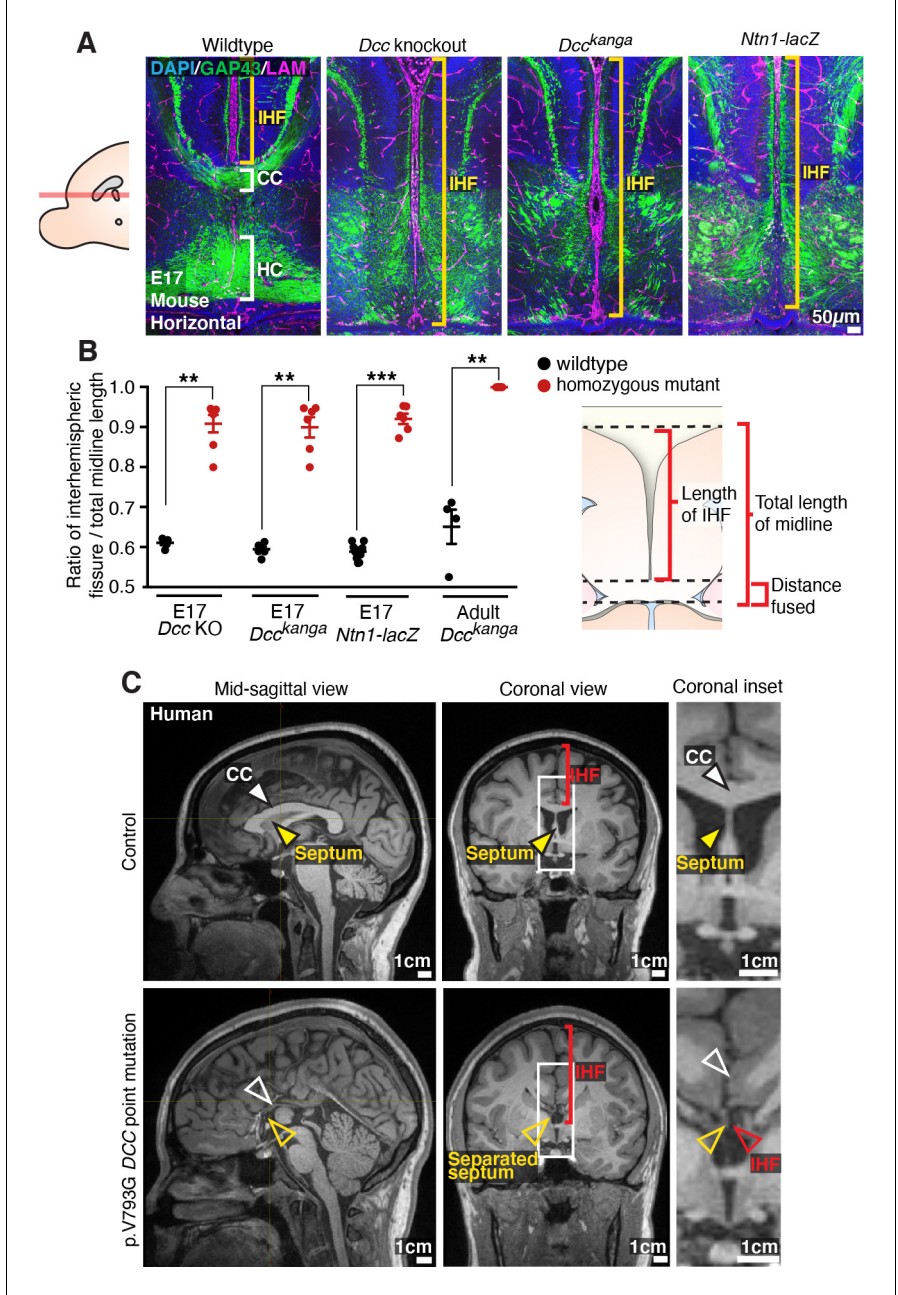

**Figure 1.** Netrin 1 (NTN1) and deleted in colorectal carcinoma (DCC) are crucial for remodelling of the interhemispheric fissure (IHF), corpus callosum (CC) and hippocampal commissure (HC) formation. (**A**) Staining for Gap43-positive axons (green) and pan-Laminin (LAM)-positive leptomeninges and basement membrane (magenta) in wildtype, *Dcc* knockout, *Dcc^{kanga}*, and *Ntn1-lacZ* mice at embryonic day (E)17 indicates midline formation or absence of the CC and HC (white brackets) and extent of the IHF (yellow brackets). (**B**) The ratio of IHF length over the total midline length with schema. (**C**) T1-weighted MR images of a control subject compared with an individual with a DCC mutation demonstrate the presence or absence of the CC (white arrowheads) and extent of the IHF (red arrowheads and brackets) within the septum (yellow arrowheads). Graph represents mean ± SEM. Statistics by Mann–Whitney test: **p<0.01, ***p<0.001. See related *Figure 1—figure supplement 1* and *Supplementary file 1*.

The online version of this article includes the following source data and figure supplement(s) for figure 1:

**Source data 1.** Ratio of interhemispheric fissure (IHF) length/total telencephalic midline length in *Dcc* and *Ntn1* mouse mutants.

**Figure supplement 1.** The interhemispheric fissure (IHF) is not remodelled in adult *Dcc^{kanga}* mice.

telencephalic hinge at E12 and E15 (*Figure 2B–D, F, H*, *Figure 2—figure supplement 1H–J*), and in MZG migrating to the IHF at E15 (*Figure 2F, H*). Furthermore, *Dcc* was expressed in Glast-positive radial glia within the septum but not in the neocortex (*Figure 2—figure supplement 1A–C*). DCC protein can be identified on Glast-positive processes of radial glia attached to the IHF (*Figure 2G*), which are adjacent to Gap43-positive axons traversing the midline region that also express DCC (*Figure 2—figure supplement 1E*). Following IHF remodelling at E17, mature Gfap-positive/Sox9-positive multipolar MZG cells (*Gobius et al., 2016*; *Sun et al., 2017*) and Glast-positive MZG cells within the telencephalic hinge continue to express DCC (*Figure 2J–L*). A comparison of DCC immunohistochemistry in wildtype and *Dcc* knockout mice confirmed that the antibody specifically recognized DCC protein within both commissural axons and MZG cells (*Figure 2—figure supplement 1K*). Importantly, we did not observe specific staining for either *Dcc* or *Ntn1* mRNA within the IHF (including the leptomeninges) at any stage analysed (*Figure 2*, *Figure 2—figure supplement 1*).

Since NTN1 is a secreted cue (*Kennedy et al., 1994*; *Lai Wing Sun et al., 2011*), we investigated which cells express NTN1, and where secreted NTN1 may be deposited, by comparing patterns of immunohistochemistry for β-galactosidase (β-gal) and NTN1 antibodies in heterozygous and homozygous *Ntn1-lacZ* mutants, in which NTN1 is fused to a β-gal and trapped in intracellular compartments (*Serafini et al., 1996*). NTN1/βgal-positive puncta were enriched in Glast-positive MZG cells in *Ntn1-lacZ* mice (*Figure 2I*). Furthermore, we identified NTN1 protein on the IHF basement membrane (*Figure 2I*, *Figure 2—figure supplement 1G*), on growing commissural axons (*Figure 2—figure supplement 1G*) and on MZG membranes in control heterozygotes, but not in *Ntn1-lacZ* homozygous mutant mice (*Figure 2I*). Therefore, MZG cells produce and secrete NTN1 that becomes deposited on the basement membrane of the IHF, on commissural axons and on MZG cell processes in the region of initial IHF remodelling (*Figure 2E*). Collectively, our results demonstrate that both *Ntn1* and *Dcc* are expressed by MZG and suggest that autocrine NTN1-DCC signalling may regulate MZG development and subsequent IHF remodelling.

## *Dcc* signalling regulates MZG cell morphology and process organization prior to IHF remodelling

Two key steps in IHF remodelling are the somal translocation of radial MZG progenitors to the IHF, and their subsequent transition into multipolar MZG cells that intercalate across the midline (*Gobius et al., 2016*). As both NTN1 and DCC are expressed by MZG, we next asked whether these molecules regulate MZG development. Immunohistochemistry for Nestin and Glast, which are markers of radial MZG, revealed distinct differences in MZG development in *Dcc^kanga* mice from E14 onwards, but not in radial MZG progenitors at E13 (*Figure 3*, *Figure 3—figure supplement 1A*). In wildtype mice, the endfeet and cell bodies of radial Glast-positive MZG cells are evenly distributed along the medial septum and adjacent to the pial surface of the IHF (*Figure 3B, D*, *Figure 4A, C*). However, in *Dcc^kanga* mutants, radial MZG accumulate at the base of the IHF (*Figure 3A–D*). Furthermore, long radial Nestin-positive MZG processes extending from the ventricular zone to the rostral-most pial surface of the IHF are noticeably absent from *Dcc^kanga* mutants, and instead, Nestin-positive *Dcc^kanga* processes cluster close to the rostral IHF pial surface and appear disorganized (*Figure 3A, C, C', E*, *Figure 3—figure supplement 1B–D*). These abnormalities were further quantified as a significant increase in fluorescence intensity of Glast staining within the base of the IHF and a concomitant decrease in the region 150–200 µm distant from the IHF base in *Dcc^kanga* mutants compared to their wildtype littermates at E14 (*Supplementary file 1*; *Figure 3B, B', G*). Just prior to IHF remodelling at E15, there was an overall decrease in Glast-positive radial MZG processes in *Dcc^kanga* mutants (*Figure 3C, H*; *Supplementary file 1*). While there was no difference in fluorescence intensity of Glast-positive radial MZG processes one day later at E16, *Dcc^kanga* MZG processes continued to display irregular morphology and failed to intercalate across the IHF (*Figure 3E, F, I*; *Supplementary file 1*). Interestingly, we identified a similar defect in the distribution of Glast-positive MZG processes in *Ntn1-lacZ* mutant mice at E15 (*Figure 3K*). These results suggest that both DCC and NTN1 are required for the correct morphology and distribution of MZG processes prior to IHF remodelling. Moreover, abnormal morphology and increased abundance of GLAST-positive and NESTIN-positive radial fibres of the dorsal glial population, known as the glial wedge, were also evident in E15 *Dcc^kanga* mice (*Figure 3C, D*), suggesting that DCC regulates the morphology and distribution of at least two midline glial populations prior to CC development.

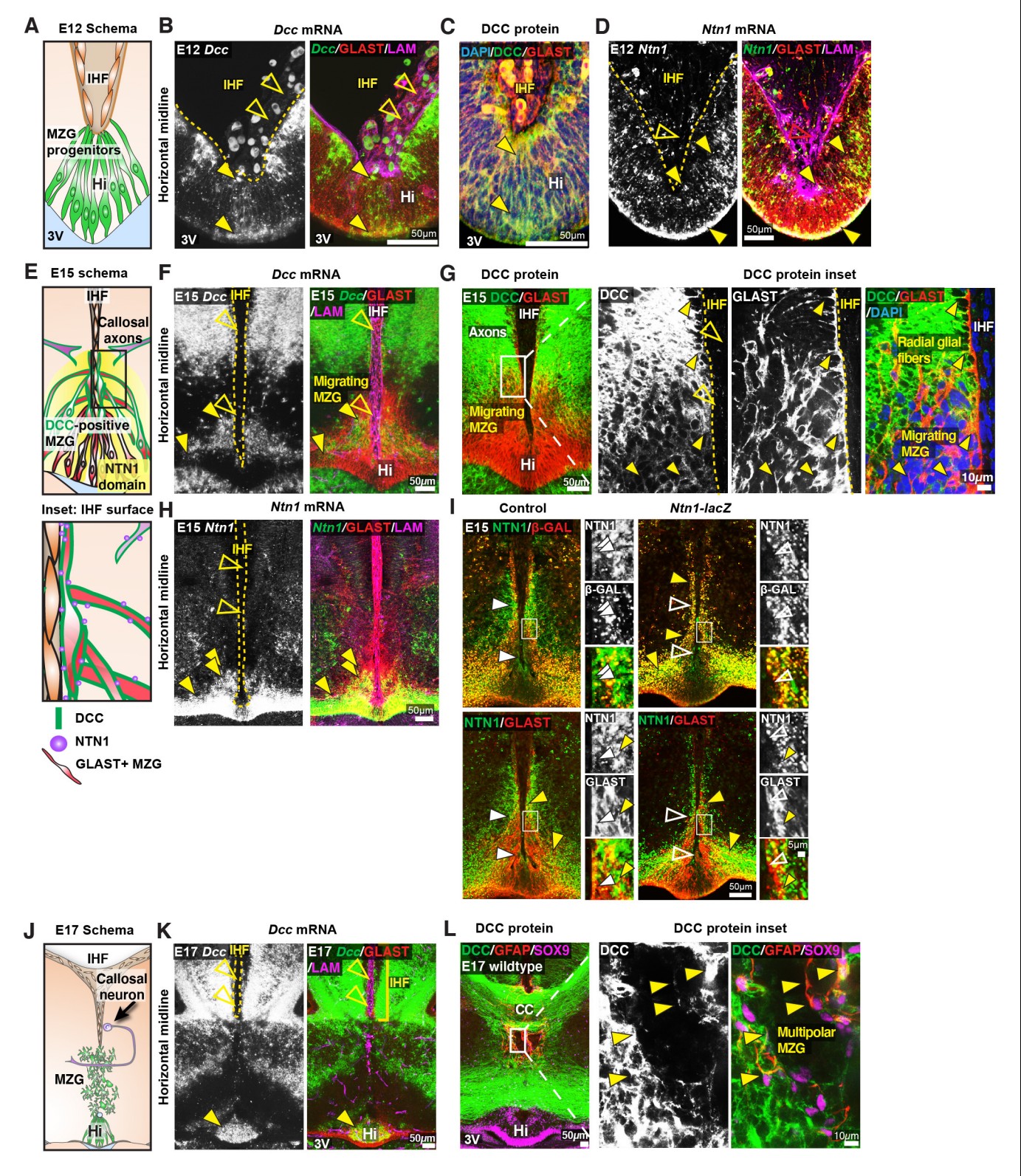

**Figure 2.** Deleted in colorectal carcinoma (DCC) and netrin 1 (NTN1) are expressed in midline zipper glia (MZG) and MZG progenitors. (**A, E, J**) Schematics depicting the cellular composition of the ventral telencephalic midline at embryonic day (E)12, E15 and E17. (**B, F, K**) *Dcc* mRNA (green), Glast-positive glia (red) and pan-Laminin (LAM)-positive leptomeninges and basement membrane (magenta) in E12, E15 and E17 wildtype mice reveal *Dcc*-positive/Glast-positive glial fibres (yellow arrowheads) and absence of *Dcc* within the interhemispheric fissure (IHF) (open yellow arrowheads). (**C, G**)
*Figure 2 continued on next page*

Figure 2 continued

DCC protein (green) and Glast protein (red) at E12 and E15 in wildtype mice reveal DCC-positive/Glast-positive glial fibres (yellow arrowheads) and absence of DCC within the IHF (open yellow arrowheads). (D, H) *Ntn1* mRNA (green), Glast (red) and pan-LAM (magenta) in E12 and E15 wildtype mice show *Ntn1*-positive/Glast-positive glial fibres (yellow arrowheads) and absence of *Ntn1* within the IHF (open yellow arrowheads). (E, inset) Schema of DCC and NTN1 expression at the E15 IHF surface, based on the results from **F–I** and *Figure 2—figure supplement 1*. (I) NTN1 (green) and Glast (red) or β-galactosidase (β-GAL; red) immunolabelling in E15 control and *Ntn1-lacZ* mice identify regions of NTN1 staining present in control heterozygotes and absent in homozygous *Ntn1-lacZ* mice (white arrowheads) and NTN1-/β-GAL-positive puncta located in Glast-positive glia (yellow arrowheads), with insets. (L) DCC protein (green), glial-specific nuclear marker SOX9 (magenta) and mature astroglial marker (GFAP) in E17 wildtype mice identify DCC-positive/GFAP-positive/SOX9-positive glia (yellow arrowheads). 3V: third ventricle; Hi: telencephalic hinge. See related *Figure 2—figure supplement 1*.

The online version of this article includes the following figure supplement(s) for figure 2:

**Figure supplement 1.** Deleted in colorectal carcinoma (DCC) is expressed in midline zipper glia (MZG).

To further characterize the defect in MZG cell distribution in *Dcc^{kanga}* mice, we then measured the maximum rostro-caudal extent to which MZG occupy the IHF pial surface and normalized this value to the total midline length from E14 to E16 (*Figure 3A–F, J*). The distribution of Nestin-positive and Glast-positive MZG along the IHF was significantly decreased at both E14 and E15 in *Dcc^{kanga}* mice compared to their wildtype littermates (*Figure 3A–D, J*; *Supplementary file 1*). The attachment of MZG processes to the IHF pial surface is therefore specifically reduced in the rostral region of the IHF prior to IHF remodelling in *Dcc^{kanga}* mice. This may impact the directed somal translocation of *Dcc^{kanga}* MZG cell bodies and their subsequent distribution along the IHF surface prior to IHF remodelling.

Next, we further investigated whether the aberrant organization of radial glial processes along the IHF in *Dcc^{kanga}* mice was due to a loss of endfoot adhesion to the IHF pial surface. There was no difference in fluorescence intensity of Nestin-positive MZG processes within 5 μm adjacent to the IHF surface between *Dcc^{kanga}* and wildtype mice, suggesting comparable attachment of radial glial endfeet to the IHF in both strains (*Supplementary file 1*; *Figure 3—figure supplement 1E, G*). This was further evidenced by the normal localization of α- and β-dystroglycan at the pial IHF surface in *Dcc^{kanga}* mice, where these molecules form crucial adhesions between radial glial endfeet and the extracellular matrix (*Myshrall et al., 2012*; *Supplementary file 1*; *Figure 3—figure supplement 1D*). Moreover, molecules that normally maintain the bipolar morphology of radial glia, such as β-catenin and N-cadherin, were also expressed normally within *Dcc^{kanga}* Nestin-positive radial glia, but adenomatous polyposis coli (APC) was instead significantly reduced in *Dcc^{kanga}* Nestin-positive radial glial endfeet (*Supplementary file 1*; *Figure 3—figure supplement 1D, E*; *Yokota et al., 2009*). APC regulates the growth and extension of basal radial glial processes and cell polarity of radial glia and migrating astrocytes (*Etienne-Manneville and Hall, 2003*; *Yokota et al., 2009*). Thus, reduced localization of APC within *Dcc^{kanga}* radial glial basal endfeet may indicate perturbed regulation of cell process growth and/or cell polarity. Therefore, *Dcc^{kanga}* Nestin-positive radial glia display reduced elongation and reduced occupation of the pial IHF surface compared to wildtype radial progenitors of MZG. Collectively, these results suggest that DCC is not required for radial MZG to adhere to the IHF, but instead regulates the morphology and organization of radial MZG processes along the pial IHF surface prior to IHF remodelling.

## DCC signalling regulates MZG somal translocation to the IHF prior to IHF remodelling

To determine if the aberrant morphology and organization of radial glial processes observed in *Dcc^{kanga}* mice affects the subsequent distribution of translocated MZG cell bodies at the IHF surface, immunohistochemistry for glial markers Sox9 and Glast was performed in E14–E16 *Dcc^{kanga}* mice. Wildtype MZG undergo substantial somal translocation to the IHF between E14 and E15 (*Gobius et al., 2016*; *Supplementary file 1*; *Figure 4A, C, G*). In contrast, *Dcc^{kanga}* mice showed reduced somal translocation to the IHF (*Supplementary file 1*; *Figure 4A, C, G*), with significantly fewer MZG cells at the IHF pial surface by E15 in *Dcc^{kanga}* compared to wildtype mice (*Supplementary file 1*; *Figure 4C*). When binned along the rostro-caudal axis, we found a significant reduction in the number of cell bodies reaching the rostral IHF pial surface in E15 *Dcc^{kanga}* mice (200–250 μm; *Supplementary file 1*; *Figure 4C, D*). Since MZG progenitors somal translocate

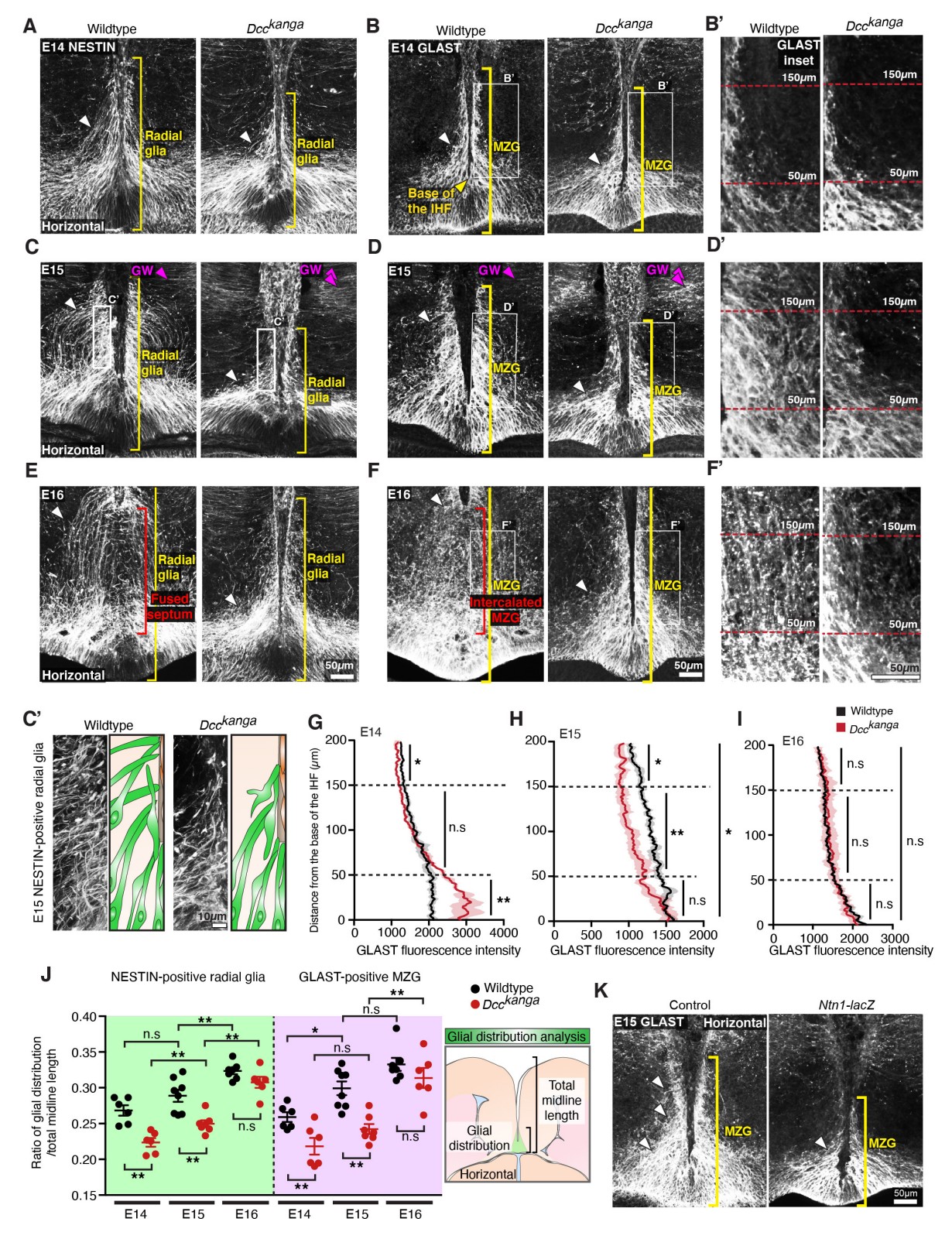

**Figure 3.** Netrin 1 (NTN1) and deleted in colorectal carcinoma (DCC) regulate midline zipper glia (MZG) morphology and spatial distribution. Nestin-positive radial glia (white; **A, C, E**) and Glast-positive glia (white; **B, D, F, K**) in embryonic day (E)14–E16 *Dcc^kanga* mice (**A–F**) and E15 *Ntn1-LacZ* mice (**K**) demonstrate the distribution of glial processes along the interhemispheric fissure (IHF) surface (yellow brackets) and lateral to the IHF (white arrowheads) with insets (**C', B', D', F'**). Radial fibres of the glial wedge (GW) are indicated with magenta arrowheads. The mean fluorescence intensity of

*Figure 3 continued on next page*

*Figure 3 continued*

Glast staining between wildtype and *Dcc^kanga* mice at E14 (G), E15 (H) and E16 (I) based on the results from (B), (D) and (F), respectively. (J) The ratio of glial distribution over total midline length, with schema, based on the results from (A) to (F). All graphs represent mean ± SEM. Statistics by Mann–Whitney test . n.s: not significant; *p<0.05, **p<0.01. See related *Figure 3—figure supplement 1* and *Supplementary file 1*.

The online version of this article includes the following source data and figure supplement(s) for figure 3:

**Source data 1.** Fluorescence intensity of GLAST and ratio of glial distribution/total midline length in *Dcc* mouse mutants.

**Figure supplement 1.** Deleted in colorectal carcinoma (DCC) is not required for endfeet attachment or molecular polarity of midline zipper glia (MZG).

**Figure supplement 1—source data 1.** Fluorescence intensity of β-dystroglycan (β-DYST), β-catenin (β-CAT), adenomatous polyposis coli (APC) and N-cadherin (N-CAD) along the interhemispheric fissure (IHF) surface in *Dcc^kanga* mice.

towards their basal process attached to the IHF (*Gobius et al., 2016*), our results suggest that the lack of radial MZG processes occupying the rostral E14 IHF surface in *Dcc^kanga* mice results in a

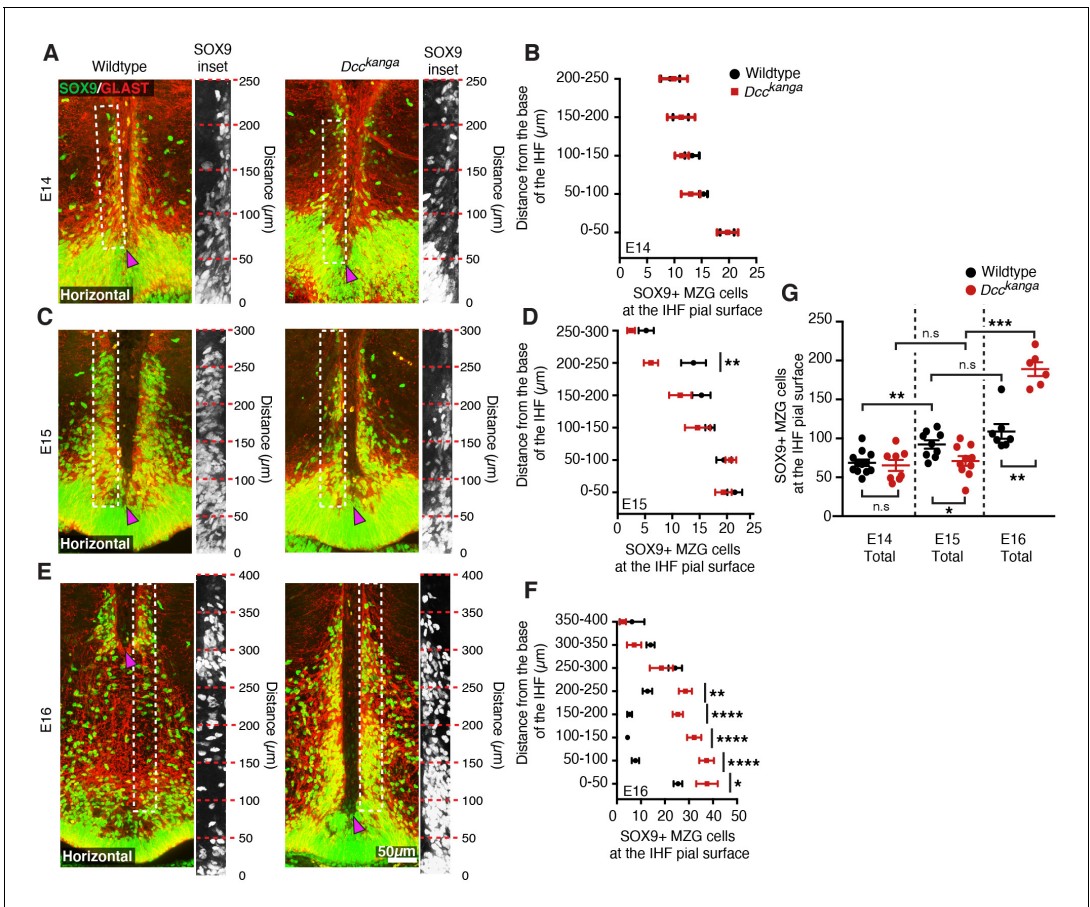

**Figure 4.** Deleted in colorectal carcinoma (DCC) regulates midline zipper glia (MZG) migration to the interhemispheric fissure (IHF) surface. (A, C, E) Nuclear glial marker SOX9 (green) and MZG marker Glast (red) in embryonic day (E)14–E16 *Dcc^kanga* mice reveal SOX9-positive/Glast-positive MZG at the pial IHF surface (boxed region and insets) above the base of the IHF (magenta arrowhead). (B, D, F, G) Quantification of SOX9-positive/Glast-positive MZG at the IHF pial surface based on the results from (A), (C) and (E). All graphs represent mean ± SEM. Statistics by Mann–Whitney test or Two-way unpaired Student's T test (G) or two-way ANOVA with post Sidak's multiple comparison test (B, D, F): *p<0.05, **p<0.01, ***p<0.001, ****p<0.0001. n.s: not significant. See related *Figure 4—figure supplement 1* and *Supplementary file 1*.

The online version of this article includes the following source data and figure supplement(s) for figure 4:

**Source data 1.** Number and distribution of SOX9-positive midline zipper glia (MZG) in *Dcc* mouse mutants.

**Figure supplement 1.** Deleted in colorectal carcinoma (DCC) does not regulate the proliferation or cell death of midline zipper glia (MZG) but regulates the formation of the indusium griseum glia and glial wedge.

**Figure supplement 1—source data 1.** Number of cells expressing 5-ethynyl-2′-deoxyuridine (EdU) and Ki67, cleaved-caspase3 and SOX9 along the interhemispheric fissure (IHF) surface in *Dcc^kanga* mice.

decrease of MZG cell bodies present in the corresponding region 1 day later. There was, however, a significant increase in MZG cell bodies present at the IHF pial surface between E15 and E16 in *Dcc^kanga* mice (*Supplementary file 1*; *Figure 4C, E, G*). This suggests that MZG migration towards the IHF is delayed but does eventually occur in *Dcc^kanga* mice, albeit after IHF remodelling would normally have been initiated. *Dcc^kanga* MZG remain adjacent to the unremodelled IHF at E16 in contrast to wildtype MZG, which are scattered along the midline where IHF remodelling has occurred and continue to expand their domain rostral and dorsal for further IHF remodelling (*Gobius et al., 2016*). Furthermore, despite DCC having been previously implicated in regulating cell proliferation and cell death (*Arakawa, 2004*; *Llambi et al., 2001*; *Mehlen et al., 1998*), cell birth-dating, differentiation and apoptosis experiments did not reveal any significant differences between the MZG of *Dcc^kanga* and wildtype mice (*Supplementary file 1*; *Figure 4—figure supplement 1*). Taken together, these results suggest that the irregular morphology and distribution of radial *Dcc^kanga* MZG processes is associated with delayed somal translocation of MZG to the IHF surface and may prevent the initiation of IHF remodelling.

Radial glia in the corticoseptal boundary detach from the pial surface and cluster their processes to form a triangular group of cells known as the glial wedge, while other radial glia in this region translocate their soma to the IHF (similar to MZG cells), where they subsequently form the indusium griseum glia (*Shu et al., 2000*; *Smith et al., 2006*). We investigated whether DCC also regulates the development of these glial populations, which were observed to be abnormal at E15 (*Figure 3C, D*) and secrete axon guidance molecules during CC formation (reviewed in *Donahoo and Richards, 2009*; *Gobius and Richards, 2011*; *Morcom and Edwards, 2016*). In *Dcc^kanga* and *Dcc* knockout mice, the glial wedge was malformed and there was a major reduction in somal translocation of Sox9-positive indusium griseum glia to the IHF surface, which subsequently prevented formation of this glial guidepost cell population (*Supplementary file 1*; *Figure 4—figure supplement 1G–I*). Thus, DCC may play a more general role in regulating the morphological maturation and migration of multiple radial astroglial populations in the developing midline, which are critical for CC formation.

## DCC signalling regulates MZG cell morphology and spatial distribution during IHF remodelling

We previously demonstrated that MZG differentiation is controlled by molecular signalling initiated by the morphogen FGF8 via the mitogen-activated protein kinase (MAPK) pathway to NFI transcription factors A and B (*Gobius et al., 2016*). Members of this signalling pathway (*Fgf8*, NFIA, NFIB and p-ERK1/2) were expressed normally in *Dcc^kanga* MZG compared to MZG in their wildtype littermates at E15 (*Figure 5—figure supplement 1B, D–F*; *Supplementary file 1*). Further, *Dcc^kanga* MZG continue to express *Mmp2* mRNA (*Figure 5—figure supplement 1C, D*; *Supplementary file 1*), which we previously demonstrated to be expressed during MZG-mediated degradation of the IHF during remodelling (*Gobius et al., 2016*).

Next, we investigated the distribution and maturation of MZG in *Ntn1* and *Dcc* mutant mice at E16 and 17 when wildtype MZG normally differentiate into multipolar astrocytes during IHF remodelling (*Gobius et al., 2016*). Immunohistochemistry for Nestin, Glast (*Figure 3F, J*) and Gfap (*Figure 5A*) demonstrated that *Dcc^kanga*, *Dcc* knockout and *Ntn1-lacZ* MZG remain attached to the caudal IHF pial surface and have not intercalated at stages equivalent to when wildtype MZG have infiltrated and remodelled the IHF (*Figure 3F, K*, *Figure 5A*; *Supplementary file 1*). DCC-deficient MZG expressed GFAP at comparable levels to wildtype MZG at E17 and demonstrated no precocious expression of GFAP at E15, similar to wildtype MZG (*Figure 5A, B*, *Figure 5—figure supplement 1A*; *Supplementary file 1*). Therefore, DCC-deficient MZG do not mature precociously prior to migration and IHF remodelling, or fail to differentiate during callosal development. However, *Ntn1-lacZ*, *Dcc^kanga* and *Dcc* knockout mice demonstrate a significant reduction of GFAP-positive glia at E17 in the region where the CC normally forms in wildtype mice (i.e., >450 µm from the third ventricle; *Figure 5A–C*; *Supplementary file 1*). Instead, MZG in these mutants remain close to the third ventricle and the majority fail to migrate. To quantify this, we normalized the level of GFAP between sections and calculated a rostro-caudal ratio of this fluorescence. We observed a significant reduction in the rostro-caudal ratio of GFAP fluorescence in *Ntn1-lacZ*, *Dcc^kanga* and *Dcc* knockout mice compared to controls (*Figure 5A–D*; *Supplementary file 1*). Since progressive migration and intercalation of MZG is required for IHF remodelling (*Gobius et al., 2016*), these results indicate

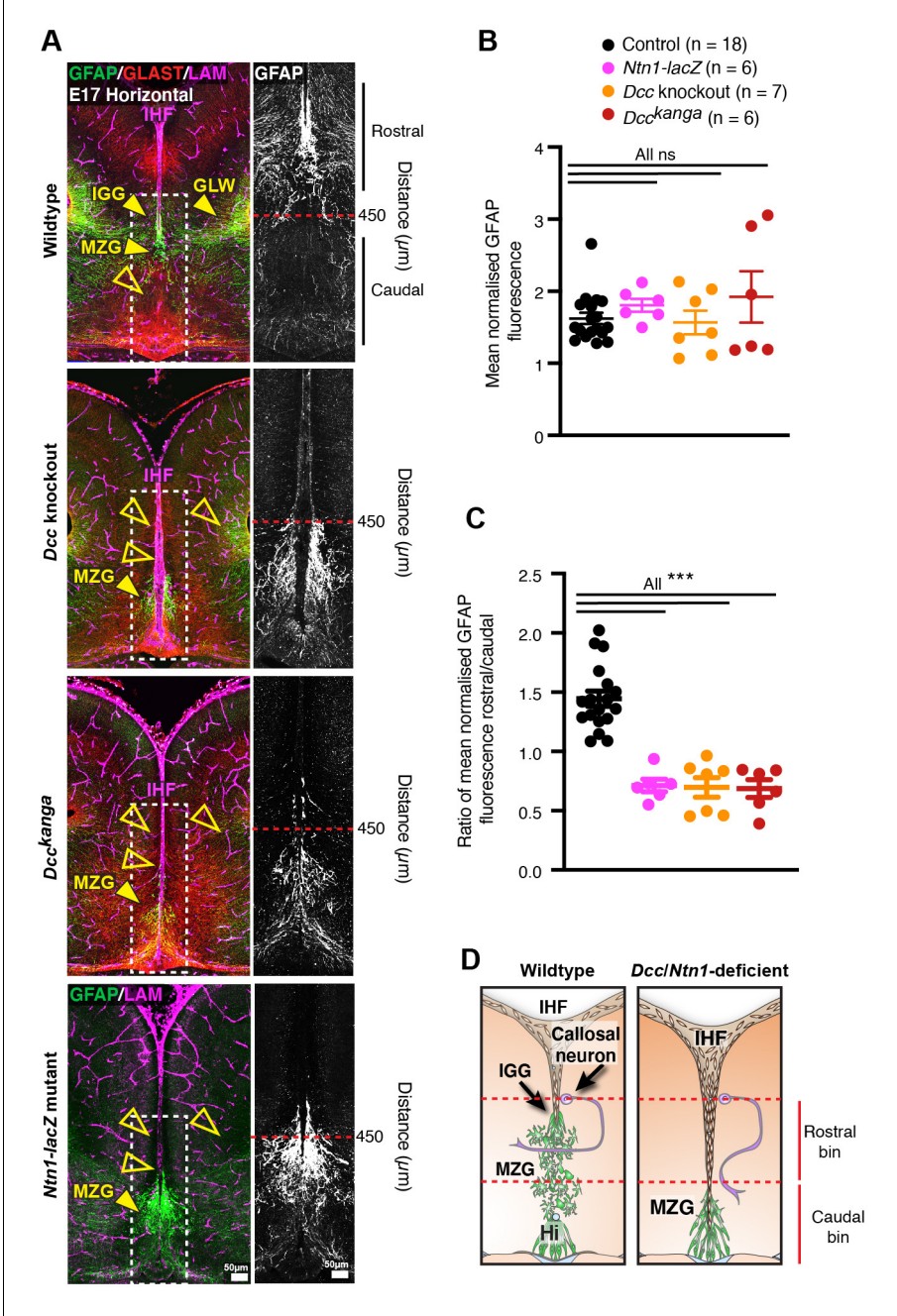

**Figure 5.** Netrin 1 (NTN1) and deleted in colorectal carcinoma (DCC) regulate midline zipper glia (MZG) organization during interhemispheric fissure (IHF) remodelling. (**A**) Gfap-positive mature astroglia (green or white in inset), Glast-positive glia (red) and pan-Laminin (LAM)-positive IHF and basement membrane (magenta) in embryonic day (E)17 wildtype $Dcc^{kanga}$, $Dcc$ knockout and $Ntn1$-LacZ mice. Yellow arrowheads indicate presence (filled) or absence (open) of midline glial populations, the MZG, indusium griseum glia (IGG) and glial wedge (GW). Fluorescence intensity of Gfap staining from insets or bins in insets (red dotted line) was quantified in (**B**) and (**C**). (**D**) Schema of MZG development, IHF remodelling and corpus callosum (CC) formation in wildtype mice and mice deficient for NTN1 or DCC. Red dotted lines indicate rostral and caudal bins that were used to calculate the ratio of GFAP fluorescence in (**C**). All graphs represent mean ± SEM. Statistics by Kruskal–Wallis test with post-hoc Dunn's multiple comparison test. \*\*\*p<0.001; ns: not significant. See related *Figure 4—figure supplement 1*, *Figure 5—figure supplement 1* and *Supplementary file 1*.

The online version of this article includes the following source data and figure supplement(s) for figure 5:

*Figure 5 continued on next page*

*Figure 5 continued*

**Source data 1.** Normalized fluorescence intensity of GFAP adjacent to the telencephalic midline in E17 *Dcc* and *Ntn1* mutant mice.
**Figure supplement 1.** Deleted in colorectal carcinoma (DCC) is not required for astroglial differentiation of midline zipper glia (MZG).
**Figure supplement 1—source data 1.** Fluorescence intensity measurements for GFAP, *Fgf8* and *Mmp-2* mRNA, and quantification of nuclear factor I (NFI)-positive/GLAST-positive cell bodies in *Dcc^kanga^* mice.

that *Ntn1* and *Dcc* affect IHF remodelling by regulating the morphology and spatial organization of both radial MZG progenitors and mature MZG, and therefore their ability to intercalate across the IHF, but not their proliferation and adhesion to the pial IHF surface.

## Variable DCC knockdown during midline development causes a spectrum of callosal phenotypes

The current and previous results from our laboratory indicate at least two distinct roles for NTN1 and DCC during CC formation: first, they act on astroglia to facilitate remodelling of the IHF, and second, they regulate the pathfinding of callosal axons to the telencephalic midline (*Fothergill et al., 2014*). To investigate these roles further, we aimed to disrupt DCC expression specifically within the progenitors of callosal neurons, sparing expression within MZG. We designed two *Dcc*-targeted CRISPR/CAS9 constructs (*Dcc*-CRISPR) and acquired a *Dcc*-targeted shRNA (*Dcc*-shRNA; *Zhang et al., 2018*) for targeted in utero electroporation into the E13 cingulate cortex in wildtype and *Dcc^kanga^* mice and successfully labelled callosal axons that reach the contralateral hemisphere by E18 (*Figure 6—figure supplement 1A*). We observed no phenotype in all experimental cases (*Figure 6—figure supplement 1A*) and instead found that these techniques failed to reduce DCC expression sufficiently; the only significant reduction in DCC protein was observed in heterozygous *Dcc^kanga^* mice electroporated with *Dcc*-shRNA (average DCC expression reduced to 93.06% compared to the non-electroporated hemisphere; *Figure 6—figure supplement 1A–C*; *Supplementary file 1*). In order to knockout DCC more robustly in the cortex, we crossed *Dcc^flox/flox^* mice (*Krimpenfort et al., 2012*) with mice carrying *Emx1^iCre^* (*Kessaris et al., 2006*) and the *tdTomato^flox_stop^* reporter allele (*Madisen et al., 2010*).

At birth, we observed a spectrum of callosal phenotypes in *Dcc* cKO mice, including complete callosal absence (4/12 mice), partial CC absence (5/12 mice) and a normal CC that was comparable to control mice, which do not express *Emx1^iCre^* (3/12 mice) based on rostral-caudal CC length across ventral, middle and dorsal horizontal sections (*Figure 6A, F*). Moreover, the HC was significantly reduced in the majority of animals and was absent in one *Dcc* cKO mouse (*Figure 6A, H*; *Supplementary file 1*). Unexpectedly, we found that the IHF was significantly retained across *Dcc* cKO mice, indicating that IHF remodelling had not been completed (*Figure 6A, D–E*; *Supplementary file 1*). The severity of callosal agenesis and HC dysgenesis was significantly correlated with the extent to which the IHF had been remodelled (*Figure 6A, J, K*, *Figure 6—figure supplement 1D–G*; *Supplementary file 1*). Complete callosal agenesis *Dcc* cKO mice demonstrated the most severe retention of the IHF, encompassing the majority of the telencephalic midline, while partial callosal agenesis and even full CC *Dcc* cKO mice demonstrated a retention of the rostral IHF (*Figure 6A, E*; *Supplementary file 1*). Moreover, in partial callosal agenesis and full CC *Dcc* cKO mice that demonstrated partial retention of the rostral IHF, callosal axons often crossed the midline more caudal in a region where the IHF had been remodelled compared to control mice (see corpus callosum remnant or CCR in *Figure 6A*). This suggests that in the absence of their normal substrate callosal axons are able to adapt and cross the midline in a region where the substrate is available. These results were reflected by a significant increase in the rostro-caudal depth of the partial or full CC in *Dcc* cKO mice (*Figure 6A, G*; *Supplementary file 1*). In *Dcc* cKO mice with complete callosal agenesis, callosal axons were unable to cross the midline and accumulated adjacent to the IHF that had not been remodelled (red arrowheads in *Figure 6A*), similar to *Dcc^kanga^* and *Dcc* knockout mice. These results demonstrate that DCC regulates the extent of IHF remodelling throughout the telencephalic midline. The retention of the IHF in these mice was unexpected; we had instead expected that reduced DCC expression in the cortex would cause callosal axon misguidance with normal

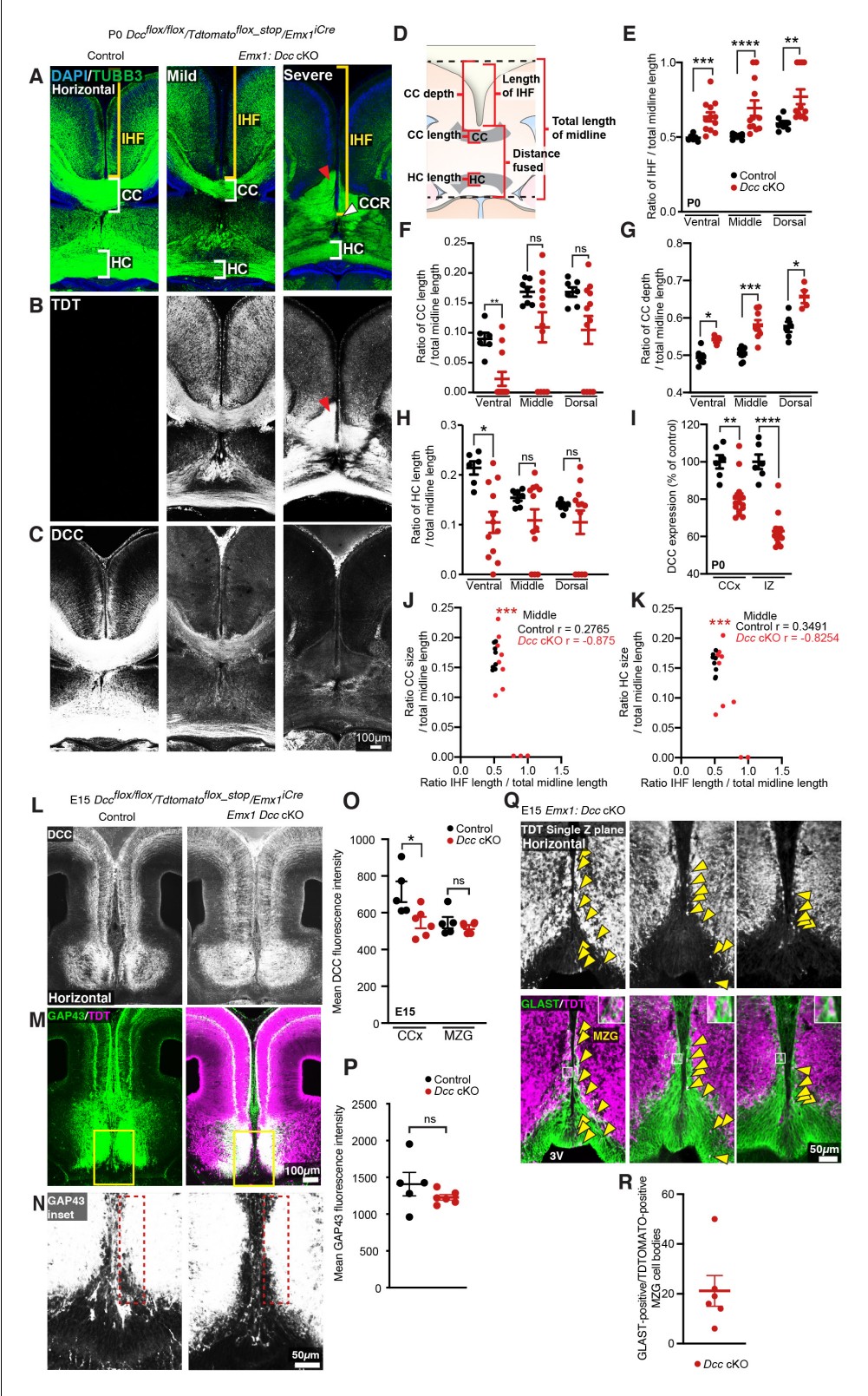

**Figure 6.** Conditional knockdown of deleted in colorectal carcinoma (DCC) within EMX1 cells causes a spectrum of callosal phenotypes. (**A**) Axonal marker TUBB3 (green), (**B**) TDT (white) or (**C**) DCC (white) in P0 *Dcc* cKO demonstrate a spectrum of callosal and interhemispheric fissure (IHF) remodelling phenotypes and a reduction in DCC expression within mice expressing *Emx1^iCre*. The corpus callosum (CC) or CC remnant (CCR) and hippocampal commissure (HC) are indicated with white brackets or white arrowheads, and the IHF is indicated with yellow brackets. Red arrowheads

*Figure 6 continued on next page*

*Figure 6 continued*

indicate axon bundles that have not crossed the midline. (D) Schema of measurements taken for quantification shown in (C–E). (E) Quantification of the ratio of IHF length normalized to total telencephalic midline length measured for P0 *Dcc* cKO mice. (F, G) Quantification of CC length (F) and depth (G) normalized to the total telencephalic midline length in P0 *Dcc* cKO mice. (H) Quantification of HC length normalized to the total telencephalic midline length in P0 *Dcc* cKO mice. (I) Quantification of DCC expression measured from the cingulate cortex (CCx) and intermediate zone (IZ) of *Dcc* cKO mice. (J, K) Scatterplots of the relationship between CC length (J) or HC length (K) normalized to total telencephalic midline length and IHF length normalized to total telencephalic midline length for middle horizontal sections of P0 *Dcc* cKO mice. Pearson r correlations are shown. (L) DCC (white), (M, N) axonal marker GAP43 (green or white, insets) and TDT (magenta) in embryonic day (E)15 *Dcc* cKO mice, with quantification of mean DCC fluorescence in (O) and quantification of mean GAP43 fluorescence within 50 μm from the IHF (dotted red lines) in (P). (Q) TDT (white or magenta) and glial marker GLAST (green) in E15 *Dcc* cKO with insets and yellow arrowheads indicating GLAST-positive/TDT-positive MZG, and quantified in (R). All graphs represent mean ± SEM. Statistics by Mann–Whitney test or unpaired t-test: *p<0.05, **p<0.01, ***p<0.001, ****p<0.0001; n.s: not significant. See related *Figure 5—figure supplement 1* and *Supplementary file 1*.

The online version of this article includes the following source data and figure supplement(s) for figure 6:

**Source data 1.** Measurements of interhemispheric fissure (IHF), corpus callosum (CC) and hippocampal commissure (HC) length and depth, deleted in colorectal carcinoma (DCC) fluorescence and GLAST-positive/TDTOMATO-positive midline zipper glia (MZG) cell bodies in *Dcc* cKO mice.
**Figure supplement 1.** *Dcc* knockdown via targeted in utero electroporation does not cause corpus callosum (CC) abnormalities.
**Figure supplement 1—source data 1.** Quantification of the ratio of deleted in colorectal carcinoma (DCC) expression between hemispheres, measurements of interhemispheric fissure (IHF) length, corpus callosum (CC) length and hippocampal commissure (HC) length, and GLAST fluorescence intensity along the IHF surface in *Dcc* cKO mice.

formation of an interhemispheric substrate. Instead, our results suggest that DCC primarily regulates the formation of the interhemispheric substrate to determine CC size in mice.

We next explored whether loss of DCC expression within callosal axons might cause prior callosal axon misguidance that could indirectly impact IHF remodelling. We found that DCC expression was significantly reduced in the cingulate cortex and adjacent intermediate zone in the majority of P0 *Dcc* cKO mice (mean expression reduced to 80.4 and 62.9%, respectively; *Figure 6A, I*; *Supplementary file 1*), and within E15 *Dcc* cKO mice (mean DCC expression in the cingulate cortex reduced to 76.6% in *Dcc* cKO; *Figure 6L, O*; *Supplementary file 1*), as expected. Surprisingly, we found that TDTOMATO-positive/GAP43-positive axons, which had reduced DCC expression, approached the interhemispheric midline in *Dcc* cKO mice, similar to their cre-negative littermates (*Figure 6M–N, P*; *Supplementary file 1*). This suggests that axons with reduced, but not entirely eliminated, DCC expression approach the midline adjacent to the IHF in a timely and spatially appropriate manner, and are unable to cross the midline in regions where the IHF is not remodelled in *Dcc* cKO mice.

Next, we investigated the recombination pattern and development of MZG in *Dcc* cKO mice. Cre activity, as measured by TDTOMATO expression, was widespread in cells throughout the telencephalic midline, including septal cells and HC axons, resulting in reduced DCC expression in multiple cell types (*Figure 6B*). Mean DCC expression within the telencephalic hinge was comparable between *Dcc* cKO mice and their cre littermates (*Figure 6L, O*; *Supplementary file 1*), but we also observed TDTOMATO-positive/GLAST-positive MZG cell bodies within the telencephalic hinge, and at the IHF surface in *Dcc* cKO mice (*Figure 6Q–R*; *Supplementary file 1*). This suggests the potential for *Dcc* knockdown in a subset of MZG cells within *Dcc* cKO mice, which may fail to intercalate across the IHF, possibly causing the IHF remodelling defect observed in P0 *Dcc* cKO mice. However, unlike *Dcc*^kanga^ mice, we were unable to find a significant population difference in the distribution of GLAST-positive MZG between *Dcc* cKO mice and their cre-negative littermates at the level of DCC knockdown observed in this model (*Figure 6—figure supplement 1H, I*; *Supplementary file 1*). Thus, the variable callosal and IHF remodelling phenotypes observed in *Dcc* cKO mice likely arise from varying degrees of DCC knockdown in these models due to the mosaic expression of *Emx1*^iCre^ within MZG. In order to further explore the role of DCC and the impact of human *DCC* mutations on the behaviour of astroglia, we next investigated the function of human *DCC* mutations using in vitro assays.

## NTN1-DCC signalling promotes cytoskeletal remodelling of astroglia and neural progenitors

Our results suggest that NTN1 and DCC may have important functions in the morphological development of radial glia more broadly. We established in vitro assays to test the function of NTN1-DCC signalling and DCC mutant receptors in regulating the morphology of astroglial-like cells. Such assays can also be used to examine human variants of DCC pathogenic mutations (see next section). To develop these assays, we employed N2A neuroblast cells, which display neural progenitor properties (*Augusti-Tocco and Sato, 1969*; *Shea et al., 1985*), as well as U251 glioma cells, which express astroglial markers and display invasive capacity (*Zhang et al., 2013*) similar to MZG cells. Importantly, endogenous DCC has previously been demonstrated to render several glioma cell lines migratory in response to a gradient of NTN1 as a chemoattractant (*Jarjour et al., 2011*). Both cell lines were transfected with either full-length DCC fused to a TDTOMATO reporter (pCAG-DCC: TDTOMATO) to express wildtype DCC or a membrane-targeted TDTOMATO reporter (pCAG-H2B-GFP-2A-Myr-TDTOMATO) as a control and stimulated with NTN1. Moreover, we transfected U251 cells with a DCC$^{kanga}$ construct (pCAG-DCC$^{kanga}$:TDTOMATO) to test whether the P3 domain was critical for NTN1-DCC signalling effects on cell morphology.

Expression of DCC:TDTOMATO in U251 cells in the absence of ligand (vehicle alone) promoted cell spreading and elongation, reflected by a significant increase in average cell area and cell perimeter, and a significant decrease in cell circularity compared to control (*Supplementary file 1*; *Figure 7A–F*). This effect was not observed following expression of the DCC$^{kanga}$ construct alone (*Supplementary file 1*; *Figure 7A–F*), suggesting that the P3 domain of DCC is critical for inducing changes in glial cell shape. We further confirmed that these morphological changes were due to the presence of the coding region of wildtype DCC by comparing to cells transfected with plasmids where DCC had been excised and only the TDTOMATO remained (pCAG-TDTOMATO; *Supplementary file 1*; *Figure 6—figure supplement 1B*, *Figure 7—figure supplement 1A*). A similar effect was observed following DCC overexpression in N2A cells, which also registered a significant increase in average cell area and cell perimeter, and decrease in cell circularity, compared to controls (*Supplementary file 1*; *Figure 7G–I*, *Figure 7—figure supplement 1G, H*), further indicating similar effects on cell morphology in glial and neural progenitor lineages. Interestingly, application of NTN1 did not affect cell shape following DCC expression in either cell line (*Supplementary file 1*; *Figure 7A–F*), suggesting that endogenous NTN1, which is known to be expressed by U251 cells (*Chen et al., 2013*), may be sufficient for activation of DCC:TDTOMATO receptors or that NTN1 is not required for this effect. To investigate this, we examined NTN1 expression in these cell lines using western blot analysis. We confirmed that both our cell lines expressed NTN1 endogenously and that transfection of DCC increased DCC levels but did not affect NTN1 expression (*Figure 7—figure supplement 1J*). No endogenous DCC was detected by western blot in either cell line (*Figure 7—figure supplement 1J*). Thus, addition of DCC induced cytoskeletal rearrangements in both N2A and U251 cells, which may involve autocrine NTN1 signalling. Typical features of DCC-expressing cells with or without bath application of NTN1 included actin-rich regions resembling filopodia, lamellipodia and membrane ruffling in U251 cells, while only filopodia were highly abundant in DCC:TDTOMATO-expressing N2A cells; all of these features were rarely observed in control cells from both cell lines (*Figure 7A, G*). No difference in cleaved-caspase 3-mediated cell apoptosis was observed following DCC expression in either cell line (*Figure 7—figure supplement 1I*). This suggests that DCC signalling does not mediate programmed cell death but rather promotes remodelling of the actin cytoskeleton in glioma and neuroblast cells in a similar manner to neurons and oligodendrocytes (*Rajasekharan et al., 2009*; *Shekarabi and Kennedy, 2002*).

## Humans with agenesis of the CC carry loss-of-function pathogenic variants in *DCC* that are unable to modulate cell shape

Having established that DCC signalling rearranges the cytoskeleton of astroglial-like cells and that the P3 domain of DCC is crucial for this function, we next investigated whether *DCC* mutant receptors from humans with dysgenesis of the CC affected this function. Site-directed mutagenesis was performed to introduce missense mutations into the pCAG-DCC:TDTOMATO expression vector in order to model mutated *DCC* receptors found in six families with previously reported cases of

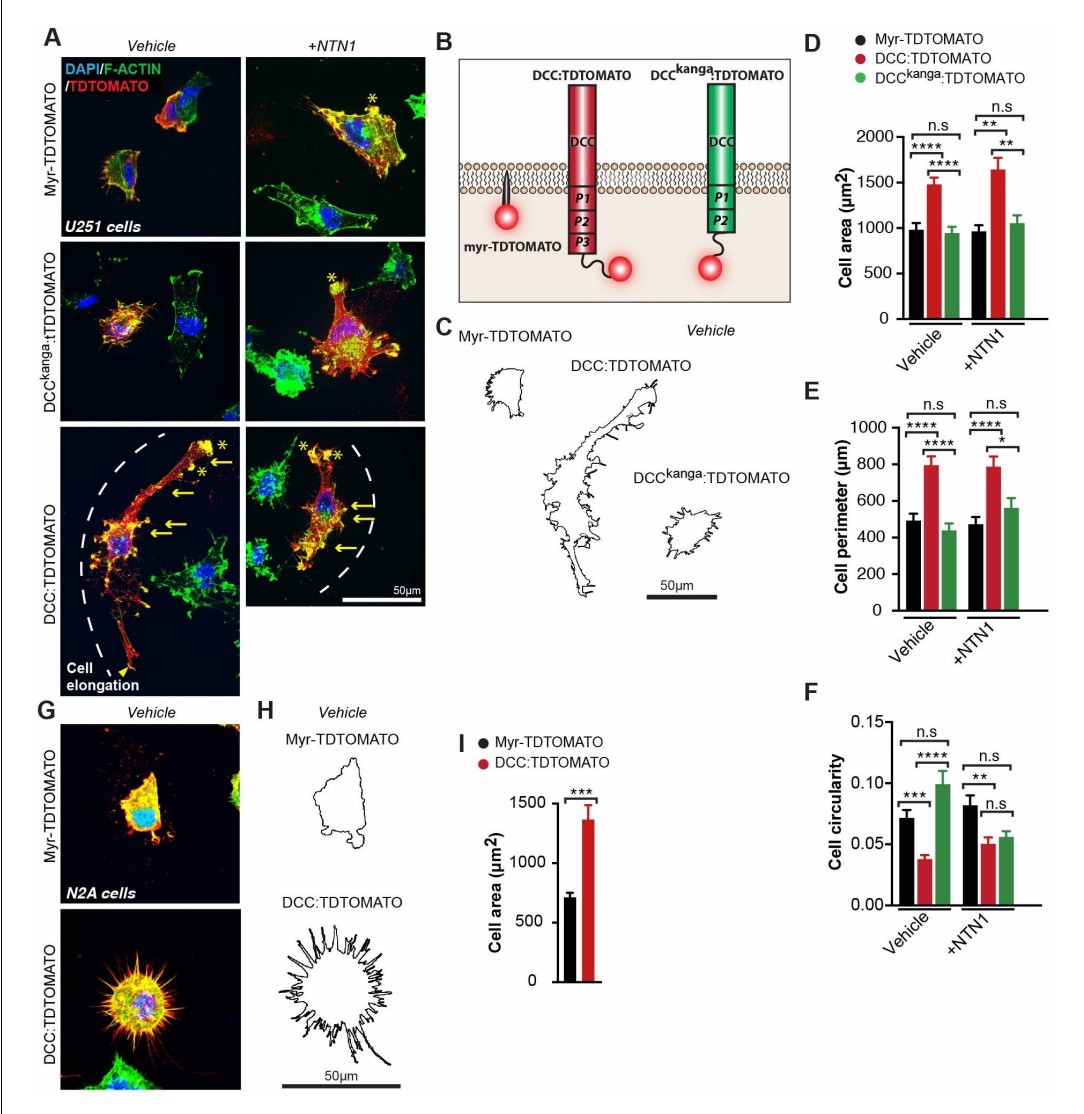

**Figure 7.** Netrin 1-deleted in colorectal carcinoma (NTN1-DCC) signalling promotes cytoskeletal remodelling of astroglia. (A, G) Representative images of U251 glioblastoma cells (A) and N2A cells (G) immunolabelled for TDTOMATO (red) and F-actin (green) following transfection with plasmids encoding Myr-TDTOMATO, DCC:TDTOMATO or DCC$^{kanga}$:TDTOMATO demonstrating the presence of actin-rich regions resembling filopodia (yellow arrows), lamellipodia (yellow arrowheads) and membrane ruffles (yellow asterisks) with/without stimulation with recombinant mouse NTN1 protein. (B) Schema of predicted structure of proteins on the cell membrane encoded by the plasmids expressed in cells from (A) and (G). (C, H) Outline of cell perimeter generated from images in (A) and (G), respectively. (D–F, I) Quantification of the area, perimeter and circularity of cells represented in (A) and (G). Graphs represent mean ± SEM. Statistics by Kruskal–Wallis test for multiple comparisons. n.s: not significant; *p<0.05, **p<0.01, ***p<0.001, ****p<0.0001. See related *Figure 6—figure supplement 1* and *Supplementary file 1*.

The online version of this article includes the following source data and figure supplement(s) for figure 7:

**Source data 1.** U251 or N2A cell area, perimeter and circularity following overexpression of DCC:TDTOMATO or myr-TDTOMATO.

**Figure supplement 1.** Mutant *DCC* receptors are expressed and trafficked normally but are unable to modulate cell shape.

**Figure supplement 1—source data 1.** U251 cell area, N2A cell perimeter and circularity and cleaved-caspase 3 expression following overexpression of DCC:TDTOMATO, DCC:TDTOMATO carrying a mutation, Myr-TDTOMATO or TDTOMATO alone.

complete or partial agenesis of the CC (p.Met743Leu, p.Val754Met, p.Ala893Thr, p.Val793Gly, p. Gly805Glu, p.Met1217Val;p.Ala1250Thr; *Marsh et al., 2017*; *Marsh et al., 2018*; *Figure 6—figure supplement 1C*, *Figure 8A*). We further included two artificial mutant receptors that were previously shown to perturb NTN1 binding and chemoattraction (p.Val848Arg, p.His857Ala; *Finci et al., 2014*). First, these mutants were transfected into HEK293T and COS-7 cells that do not endogenously

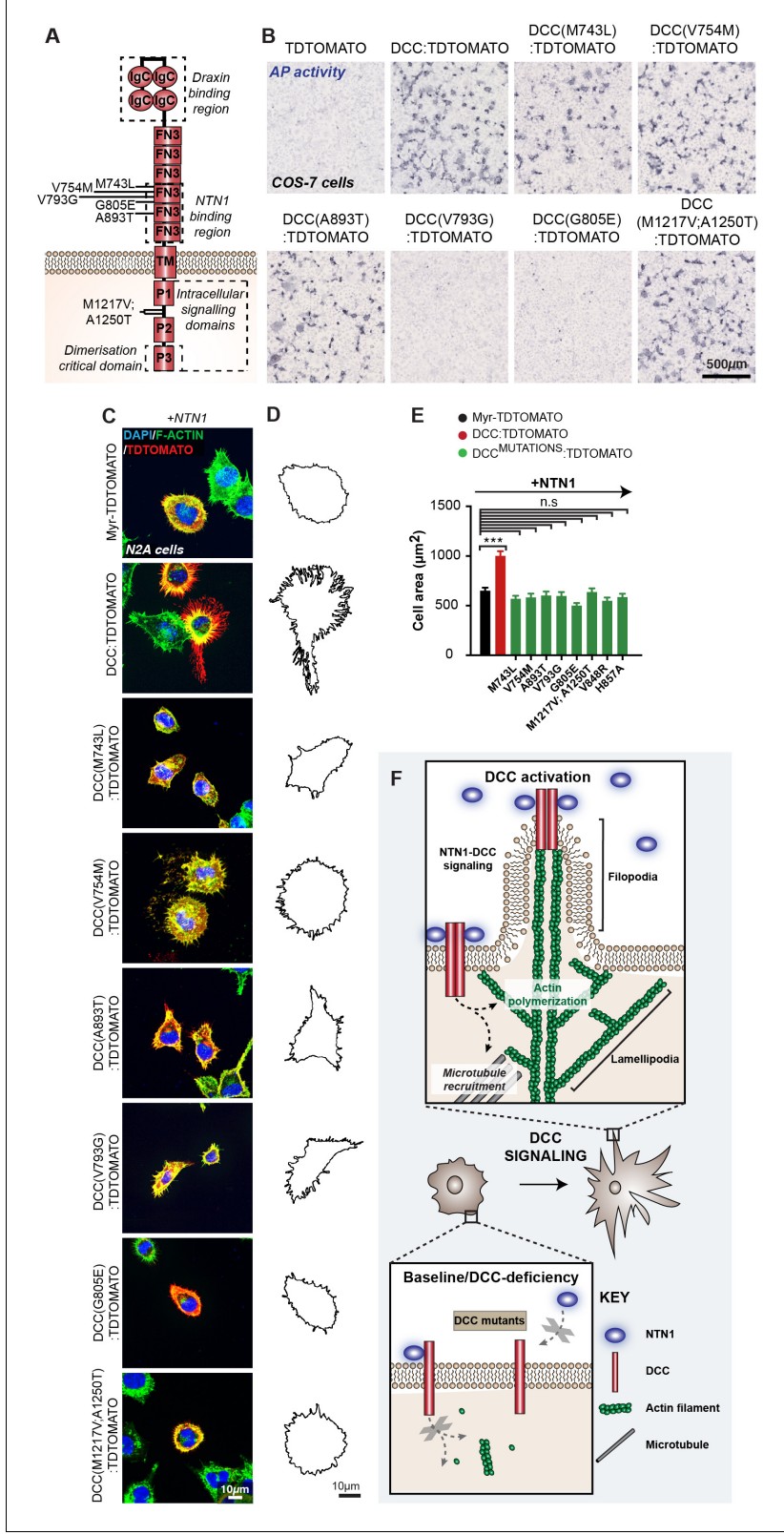

**Figure 8.** *DCC* mutations associated with human callosal agenesis are unable to modulate cell shape and show varied netrin 1 (NTN1) binding. (**A**) Schema of transmembrane receptor deleted in colorectal carcinoma (DCC) and its structural domains. Lines indicate the position of altered residues from missense *DCC* pathogenic variants identified in human individuals with corpus callosum (CC) abnormalities. FN3: fibronectin type III-like domain;

*Figure 8 continued on next page*

*Figure 8 continued*

IgC: immunoglobulin like type C domain; TM: transmembrane domain; p: P motif. (**B**) Colorimetric detection of alkaline phosphatase activity in COS-7 cells transfected with plasmids encoding TDTOMATO, DCC:TDTOMATO and mutant DCC:TDTOMATO constructs and incubated with a NTN1 alkaline phosphatase fusion protein. (**C**) Representative images of N2A cells immunolabelled for TDTOMATO (red) and F-actin (green) following transfection with plasmids encoding Myr-TDTOMATO, DCC:TDTOMATO or DCC:TDTOMATO carrying missense mutations and stimulated with recombinant mouse NTN1 protein. (**D**) Outline of cell perimeter generated from images in (**B**). (**E**) Quantification of the area of cells represented in (**B**). Graph represents mean ± SEM. Statistics by Kruskal–Wallis test for multiple comparisons. n.s: not significant, ***p<0.001. (**F**) Schema of model for DCC-mediated changes in cell shape: activation of DCC by NTN1 induces dimerization of the receptor and recruits intracellular signalling effectors to regulate actin polymerization for filopodia and lamellipodia formation, and to regulate microtubule dynamics to promote membrane protrusions. Mutations that affect DCC signalling prevent DCC-mediated changes in cell shape. See related *Figure 6—figure supplement 1* and *Supplementary file 1*. The online version of this article includes the following source data for figure 8:

**Source data 1.** N2A cell area following overexpression of DCC:TDTOMATO, DCC:TDTOMATO carrying a mutation or myr-TDTOMATO.

express DCC (*Chen et al., 2013*; *Shekarabi and Kennedy, 2002*). Immunoblotting and immunohistochemistry performed without cell permeabilization revealed that all mutant DCC receptors were appropriately expressed and localized to the cell membrane (*Gad et al., 2000*; *Figure 7—figure supplement 1E*). Using a previously established in vitro binding assay (*Müller and Soares, 2006*; *Zelina et al., 2014*), we discovered that DCC mutant proteins with altered residues located at the NTN1 binding interface (p.V793G and p.G805E) were unable to bind NTN1 (*Figure 8B*), while all other receptors with altered residues lying outside of the NTN1 binding interface still bound NTN1 (p.M743L, p.V754M, p.A893T and p.M1217;A1250T; *Figure 8B*). Surprisingly, all eight mutant *DCC* receptors were unable to modulate cell morphology in the presence of NTN1 (*Figure 8C–E*; *Supplementary file 1*). Collectively, our results suggest a model whereby mutations that affect the ability for DCC to regulate cell shape (*Figure 8F*) are likely to cause callosal agenesis through perturbed MZG migration and IHF remodelling.

## Discussion

Genes that encode axon guidance molecules frequently cause callosal dysgenesis when knocked out in mice (*Edwards et al., 2014*). This has led to the prevalent view that callosal dysgenesis in these mice might be primarily due to defects in callosal axon guidance towards and across the midline. Here, we identified a novel function for the classical axon guidance genes NTN1 and DCC in regulating the morphology of midline astroglia for IHF remodelling prior to CC and HC formation. Importantly, normal astroglial development and IHF remodelling are critical processes that precede and are necessary for subsequent CC axon guidance across the interhemispheric midline (*Gobius et al., 2016*). We find that defects in IHF remodelling are consistently associated with dysgenesis of the CC and HC in mice and humans with pathogenic variants in *Ntn1* or *Dcc*.

Our in vitro assays and analysis of mouse and human cell morphology indicate that the cytoskeletal remodelling function of NTN1-DCC signalling is likely to be crucial for MZG development, IHF remodelling and subsequent CC formation. The timely differentiation and appropriate distribution of MZG cells at the IHF surface is required for their intercalation and IHF remodelling function (*Gobius et al., 2016*). Our data suggest a model whereby failed IHF remodelling associated with mutations in *Ntn1* and *Dcc* occurs due to delayed astroglial migration to the IHF as a consequence of perturbed process extension and organization of MZG precursors. Notably, no medial extension of MZG processes across the basement membrane or perforations in the IHF to allow glia from each hemisphere to interact and intercalate were observed in *Ntn1* or *Dcc* mutant mice at any developmental stage examined. This suggests that NTN1-DCC signalling might also be required for MZG intercalation and removal of the intervening leptomeninges. The DCC homolog UNC-40 is known to facilitate formation of a polarized actin-rich cell protrusion in the *Caenorhabditis elegans* anchor cell, which breaches the basement membrane rich in UNC-6 (NTN1 homolog), enabling the cell to invade the vulval epithelium (*Hagedorn et al., 2013*; *Ziel et al., 2009*). DCC may perform a similar function in MZG by engaging secreted NTN1, which we found to be localized at the IHF basement

membrane in agreement with the localization of radial-glial-derived NTN1 in the spinal cord (*Varadarajan et al., 2017*) and preferentially polarizing actin remodelling and process extension towards the IHF during MZG intercalation. Moreover, callosal axons that also rely on DCC-mediated cytoskeletal remodelling for growth and guidance may non-cell-autonomously influence the final stages of MZG development via a secreted cue or contact-dependent mechanism. Further dissecting this would ideally involve even greater precision in complete and cell-type-specific knockout of DCC and NTN1 since knockdown in a subset of cells or merely lowering the expression level was insufficient to induce a consistent phenotype.

Notably, we find that the P3 domain-dependent functions of DCC may be required for astroglial development and IHF remodelling. These functions include receptor dimerization, interaction with the co-receptor ROBO1 or interaction with effectors FAK, MYO10 and TUBB3 (*Fothergill et al., 2014*; *Li et al., 2004*; *Qu et al., 2013*; *Stein and Tessier-Lavigne, 2001*; *Wei et al., 2011*; *Xu et al., 2018*). Accordingly, mice deficient in *Robo1, Fak* and *Tubb3*, as well as their signalling effectors *Cdc42, Fyn*, *Enah* and *Mena,* which normally act downstream of DCC to regulate the cell cytoskeleton, all display dysgenesis of the CC (*Andrews et al., 2006*; *Beggs et al., 2003*; *Goto et al., 2008*; *Menzies et al., 2004*; *Tischfield et al., 2010*; *Yokota et al., 2010*). Similarly, astroglial cells remodel their cytoskeleton to transition from a bipolar to multipolar morphology, and this process is known to involve the intracellular DCC effectors CDC42, RAC1, RHOA, N-WASP and EZRIN (*Abe and Misawa, 2003*; *Antoine-Bertrand et al., 2011*; *Derouiche and Frotscher, 2001*; *Lavialle et al., 2011*; *Murk et al., 2013*; *Racchetti et al., 2012*; *Shekarabi et al., 2005*; *Zeug et al., 2018*). Whether these molecules serve as downstream effectors of DCC to influence astroglial development and IHF remodelling during CC formation is an interesting question for future research.

In addition to NTN1 and DCC, as shown here, mice lacking the axon guidance molecules DRAXIN, ENAH, SLIT2, SLIT3 and RTN4R have previously been reported to have incomplete IHF remodelling and disrupted midline glial development associated with callosal dysgenesis (*Menzies et al., 2004*; *Morcom et al., 2020*; *Unni et al., 2012*; *Yoo et al., 2017*). Taken together, those studies and ours suggest that other axon guidance genes may play similar roles in astroglial development and IHF remodelling during CC formation. Additional candidate axon guidance molecules that may regulate IHF remodelling include EPHB1, EFNB3, GAP43, HS6ST1, HS2ST1, ROBO1 and VASP since mouse mutants lacking these molecules display disrupted midline glial development and callosal dysgenesis (*Andrews et al., 2006*; *Conway et al., 2011*; *Mendes et al., 2006*; *Menzies et al., 2004*; *Shen et al., 2004*; *Unni et al., 2012*). Additional molecules of interest are EFNB1, EFNB3, EPHB2 and EPHA4 since these are expressed by MZG (*Mendes et al., 2006*).

In summary, we have demonstrated that rather than solely regulating axon guidance during telencephalic commissure formation, *Dcc* and *Ntn1* are critical genes required for IHF remodelling. Moreover, our study provides a novel role for axon guidance receptor DCC in regulating astroglial morphology, organization and migration. Exemplified by *Ntn1* and *Dcc,* our study provides support for widespread consideration of astroglial development and IHF remodelling as possible underlying mechanisms regulated by these and other classically regarded 'axon guidance genes' during CC formation.

## Materials and methods

### Experimental models and subject details

#### Animals

*Dcc^flox/flox* (*Krimpenfort et al., 2012*), *Dcc* knockout (*Fazeli et al., 1997*), *Dcc^kanga* (*Finger et al., 2002*), *Emx1^iCre* (*Kessaris et al., 2006*), *Ntn1-lacZ* (*Serafini et al., 1996*) and *tdTomato^flox_stop* (*Madisen et al., 2010*) mice on the C57BL/6J background and CD1 wildtype mice were bred at the University of Queensland. Prior approval for all breeding and experiments was obtained from the University of Queensland Animal Ethics Committee. Male and female mice were placed together overnight, and the following morning was designated as E0 if a vaginal plug was detected. *Dcc* knockout and *Dcc^kanga* mice were genotyped by PCR, and *Ntn1-lacZ* mice were tested for the presence of the *LacZ* gene and deemed homozygous if the β-galactosidase enzyme was trapped intracellularly, as previously described (*Fazeli et al., 1997*; *Finger et al., 2002*; *Fothergill et al., 2014*;

*Krimpenfort et al., 2012*; *Serafini et al., 1996*). Dcc<sup>flox/flox</sup> mice were genotyped by the Australian Equine Genetics Research Centre at the University of Queensland.

## Human subjects

Ethics for human experimentation was acquired by local ethics committees at the University of Queensland (Australia), the Royal Children's hospital (Australia) and UCSF Benioff Children's Hospital (USA). Genetic studies were performed previously (*Marsh et al., 2017*). Structural MR images were acquired as previously described (*Marsh et al., 2017*). In our study, we analysed the brain phenotype of affected individuals in family 2 (carrying *DCC* p.Val793Gly) and family 9 (carrying *DCC* p. Met1217Val;p.Ala1250Thr in cis) from our previous study.

## Method details

### Cell birth-dating and tissue collection

For cell birth-dating studies, 5-ethynyl-2′-deoxyuridine (EdU; 5 mg per kg body weight, Invitrogen) dissolved in sterile phosphate buffer solution (PBS) was injected into the intraperitoneal cavity of awake pregnant dams. Brains were fixed via transcardial perfusion or immersion fixation with 4% paraformaldehyde (PFA).

### Cell lines

HEK293 cells (from ATCC CRL-1573, not authenticated, free of mycoplasma contamination) were used to express alkaline phosphatase-conjugated NTN1 (NTN1-AP) in the supernatant of COS-7 cell culture. Although this cell line is commonly misidentified, this did not affect the conclusion of the binding assay done in COS-7 cells. U251 cells were obtained as U-373MG (RRID:CVCL_2219) but subsequently identified as U-251 via PCR-based short tandem repeat profiling. All cell lines were routinely tested for mycoplasma to ensure that cell lines were free of mycoplasma contamination. See the Key resources table (Appendix 1—key resources table) for more information.

### Cell culture

All cell lines were cultured at 37°C within a humidified atmosphere containing 5% $CO_2$ and immersed in Dulbecco's Modified Eagle's Medium medium (Invitrogen or HyClone), supplemented with 10% fetal bovine serum. U251 cells were plated on poly-d-lysine-coated coverslips (via submersion in 0.05 mg/mL solution, Sigma-Aldrich) at 10% confluence 24 hr prior to transfection. The pCAG-TDTO-MATO, pCAG-H2B-GFP-2A-MyrTDTOMATO, pCAG-DCC:TDTOMATO and pCAG-DCC<sup>kanga</sup>:TDTO-MATO plasmids (1 µg) were transfected into the plated U251 cells using FuGENE 6 (Promega) in Opti-MEM (Gibco, Life Technologies). Cells were then grown for 20 hr and either fixed with 4% PFA/ 4% sucrose or stimulated with ligand. Since 100 ng/mL of recombinant NTN1 is sufficient to induce morphological changes in primary oligodendrocyte precursor cells (*Rajasekharan et al., 2009*), 200 ng of recombinant mouse NTN1 protein (R&D Systems) was diluted in sterile PBS and added to cultures within 2 mL media. When ligand was added, cells were grown for a further 12 hr before fixation with 4% PFA/4% sucrose. N2A cells were cultured and transfected as outlined for the U251 cells except that after NTN1 stimulation the cells were cultured for only 8 hr before fixation. The pCAG-DCC:TDTOMATO wildtype and missense mutant receptor constructs (1.764 µg) were also transfected into HEK293T cells cultured on acid-washed coverslips using FuGENE HD (Promega) in Opti-mem (Gibco, Life Technologies). After 24 hr, cells were fixed with 4% PFA/4% sucrose.

### NTN1-binding assay

Supernatant containing alkaline phosphatase-conjugated *NTN1* (*NTN1-AP*) was generated from expression in HEK293T cells as previously described (*Zelina et al., 2014*). The pCAG-DCC:TDTO-MATO wildtype and missense mutant receptor constructs (0.2 µg) were transfected into COS-7 cells using Lipofectamine 2000 (Invitrogen). After 48 hr, cells were incubated with *NTN1*-AP supernatant (1:50) for 90 min at room temperature. Cells were washed and *NTN1*-binding activity was determined using colorimetric detection as previously described (*Zelina et al., 2014*).

## Western blot

Whole-cell protein extracts were prepared from N2A and U251 cells, 20 hr after transfection with pCAG-DCC:TDTOMATO and pCAG-myr-TDTOMATO constructs (1 µg) as previously described (*Bunt et al., 2010*). Moreover, COS-7 protein extracts were prepared 48 hr after transfection with the pCAG-DCC:TDTOMATO wildtype and missense mutant receptor constructs (0.2 µg). Western blots were performed to detect mouse DCC expression levels using a goat polyclonal anti-DCC antibody (1:200 COS-7 or 1:800, sc-6535, Santa Cruz Biotechnology) and mouse NTN1 using a goat polyclonal antibody (1:500 U251 or 1:1000 N2A, AF1109, R&D Systems). GADPH was used as a loading control and detected using rabbit monoclonal anti-GADPH antibodies (1:2000, 2118, Cell Signaling Technology for COS-7; 1:1000, IMG-5143A, IMGENEX for N2A and U251).

## Immunohistochemistry

Brain sections were processed for standard fluorescence immunohistochemistry as previously described (*Moldrich et al., 2010*) with the following minor modifications: all sections were post-fixed on slides with 4% PFA and then subjected to antigen retrieval (125°C for 4 min at 15 psi in sodium citrate buffer) prior to incubation with primary antibodies. Alexa Fluor IgG (Invitrogen), horseradish peroxidase-conjugated (Millipore) or biotinylated (Jackson Laboratories) secondary antibodies, used in conjunction with Alexa Fluor 647-conjugated Streptavidin (Invitrogen) amplification, were used according to the manufacturer's instructions. EdU labelling was performed using the Click-iT EdU Alexa Fluor 488 Imaging Kit (Invitrogen). Cell nuclei were labelled using 4′,6-diamidino-2-phenylindole, dihydrochloride (DAPI, Invitrogen) and coverslipped using ProLong Gold anti-fade reagent (Invitrogen) as mounting media. Primary antibodies used for immunohistochemistry were rabbit anti-APC (1:250, ab15270, Abcam), mouse anti-α-dystroglycan (1:250, clone IIH6C4, 05-593, Merck), rabbit anti-β-catenin (1:500, 9562, Cell Signaling Technology), mouse anti-β-dystroglycan (1:50, MANDAG2, 7D11, Developmental Studies Hybridoma Bank), chicken anti β-galactosidase (1:500, ab9361, Abcam), rabbit anti-cleaved-caspase 3 (1:500, 9661, Cell Signaling Technology), goat anti-DCC (1:500, sc-6535, Santa Cruz Biotechnology), mouse anti-Gap43 (1:500; MAB347, Millipore), mouse anti-Gfap (1:500; MAB3402, Millipore), rabbit anti-Gfap (1:500; Z0334, Dako), mouse anti-Glast (or Eaat1; 1:500; ab49643, Abcam), rabbit anti-Glast (or Eaat1; 1:250; ab416, Abcam), mouse anti-Ki67 (1:500; 550609, BD Pharmingen), chicken anti-Laminin (1:500; LS-C96142, LSBio), rabbit anti-Laminin (1:500; L9393, Sigma), mouse anti-N-cadherin (CDH2; 1:250, 610921, BD Biosciences), rat anti-Nestin (NES; 1:50, AB 2235915, DSHB), chicken anti-Nestin (1:1000, ab134017, Abcam), goat anti-NTN1 (1:500; AF1109, R&D Systems), mouse anti-neurofilament (1:500; MAB1621, Millipore), rabbit anti-Nfia (1:500; ARP32714, Aviva Systems Biology), rabbit anti-Nfib (1:500; HPA003956, Sigma), rabbit anti-neuronal-specific-ßIII-tubulin (1:500; ab18207, Abcam), rabbit anti-phospho p44/42 Mapk (or Erk1/2; 1:250; 9101, Cell Signaling Technology), rabbit anti-SOX9 (1:500, AB5535, Merck) and goat anti-TDTOMATO (1:500, ab8181-200, Sicgen). For actin staining, Alexa Fluor-conjugated phalloidin (A22287, Thermo Fisher Scientific) was incubated on tissue for 30 min in the dark as per the manufacturer's instructions prior to the addition of primary antibodies. Immunohistochemistry was performed in a similar manner for cultured cells, with the following minor exceptions: HEK293T cells expressing wildtype and mutant pCAG-DCC:TDTOMATO constructs were not permeabilized to confirm exogenous DCC receptor localization to the plasma membrane.

## In situ hybridization

In situ hybridization was performed as previously described (*Moldrich et al., 2010*), with the following minor modifications: Fast red (Roche) was applied to detect probes with fluorescence. The *Fgf8* cDNA plasmid was a kind gift from Gail Martin, University of California, San Francisco. The *Ntn1* cDNA plasmid was provided by the Cooper lab. The *Mmp2* cDNA plasmid was generated by the Richards lab with the following primers: forward 5′-GAAGTATGGATTCTGTCCCGAG-3′ and reverse 5′-GCATCTACTTGCTGGACATCAG-3′. The *Dcc* cDNA plasmid was generated by the Richards lab with the following primers, courtesy of the Allen Developing Brain Atlas: forward 5′- ATGGTGAC-CAAGAACAGAAGGT-3′ and reverse 5′-AATCACTGCTACAATCACCACG-3′.

## Plasmid expression constructs for cell culture and in utero electroporation

A TDTOMATO fluorophore (Clontech) was subcloned into a *pCAG* backbone to generate the pCAG-TDTOMATO plasmid. pCAG-H2B-GFP-2A-MyrTDTOMATO was provided by Arnold Kriegstein (University of California San Francisco). The *Dcc*-shRNA construct was provided by Xiong Zhiqi (Chinese Academy of Sciences, Shanghai; shRNA 1355 in Zhang et al., 2018). The *Dcc*-CRISPR nickase constructs were designed using the ATUM tool and obtained from ATUM to target *Dcc* exon 2 (*Dcc*-CRISPR 1, targeting chr18:71954969–71955009) and *Dcc* exon 3 (*Dcc*-CRISPR 2, targeting chr18:71826146–71826092). *Dcc*-CRISPR 1 had the maximum target score across the whole DCC coding sequence, while *Dcc*-CRISPR 1 had the maximum target score within exon 3 only.

To generate the pCAG-DCC:tdTomato plasmid, DCC:TDTOMATO (pmDCC:TDTOMATO; provided by Erik Dent, University of Wisconsin-Madison) was subcloned into the pCag-DsRed2 plasmid (Addgene, 15777, Cambridge, MA) by excising DsRed2.

For site-directed mutagenesis, the QuickChange II Site-Directed Mutagenesis Kit (Stratagene, catalogue #200524) was used in accordance with the manufacturer's instructions. The following primer pairs were used for site-directed mutagenesis:

p.Met743Leu: forward 5′-GAGGAGGTGTCCAACTCAAGATGATACAGTTTGTCTG-3′, reverse 5′-CAGACAAACTGTATCATCTTGAGTTGGACACCTCCTC-3′.

p.Val754Met: forward 5′-TAATATAGCCTCTCACCATGATGTTTGGGTTGAGAGG-3′, reverse 5′-CCTCTCAACCCAAACATCATGGTGAGAGGCTATATTA-3′.

p.Ala893Thr: forward 5′-ACTTGTACTTGGTACTGGCAGAAAAGCTGGTCCT-3′, reverse 5′-AGGACCAGCTTTTCTGCCAGTACCAAGTACAAGT-3′.

p.Val793Gl: forward 5′-ACTAGAGTCGAGTTCTCATTATGGAATCTCCTTAAAAGCTTTCAAC-3′, reverse 5′- GTTGAAAGCTTTTAAGGAGATTCCATAATGAGAACTCGACTCTAGT-3′.

p.Gly805Glu: forward 5′-CACTTTCGTAGAGAGGGACCTCTTCTCCGGCATTGTTGAA-3′, reverse 5′- TTCAACAATGCCGGAGAAGAGGTCCCTCTCTACGAAAGTG-3′.

p.Met1217Val;p.Ala1250Thr: forward 1 5′-GTTCCAAAGTGGACACGGAGCTGCCTGCGTC-3′, reverse 1 5′-GACGCAGGCAGCTCCGTGTCCACTTTGGAAC-3′, forward 2 5′-GTACAGGGATGG TACTCACAACAGCAGGATTACTGG-3′, reverse 2 5′-CCAGTAATCCTGCTGTTGTGAGTACCATCCC TGTAC-3′.

p.Val848Arg: forward 5′-CAGCCTGTACACCTCTTGGTGGGAGCATGGGGG-3′, reverse 5′-CCCCCATGCTCCCACCAAGAGGTGTACAGGCTG-3′.

p.His857Ala: forward 5′-ACCCTCACAGCCTCAGCGGTAAGAGCCACAGC-3′, reverse 5′-GCTG TGGCTCTTACCGCTGAGGCTGTGAGGG-3′.

p.del-P3(Kanga): forward 5′-CCACAGAGGATCCAGCCAGTGGAGATCCACC-3′, reverse 5′-GG TGGATCTCCACTGGCTGGATCCTCTGTGG-3′.

## In utero electroporation

In utero electroporation was performed as previously described (Suárez et al., 2014b). Briefly, 2 μg/ μL of *Dcc*-shRNA or *Dcc*-CRISPR were combined with 0.5 μg/μL TDTOMATO and 0.0025% Fast Green dye, and then microinjected into the lateral ventricles of E13 *Dcc*$^{kanga}$ embryos. 5 × 35 V square wave pulses separated by 100 ms were administered with 3 mm paddles over the head of the embryo to direct the DNA into the cingulate cortex. Embryos were collected at E18 for analysis.

## Image acquisition

Confocal images were acquired as either single 0.4–0.9 μm optical sections or multiple image projections of ~15–20 μm thick z-stacks using either an inverted Zeiss Axio-Observer fitted with a W1 Yokogawa spinning disk module, Hamamatsu Flash4.0 sCMOS camera and Slidebook 6 software or an inverted Nikon TiE fitted with a Spectral Applied Research Diskovery spinning disk module, Hamamatsu Flash4.0 sCMOS camera and Nikon NIS software. Alternatively, for images of HEK293T cells, a LSM 780 confocal microscope was used. For imaging of NTN1-AP binding, a NanoZoomer 2.0-HT whole slide imager was used in conjunction with Hamamatsu (NDP_Viewer) software. For wide-field imaging of U251 and N2A cells stained for cleaved-caspase 3, Zen software (Carl Zeiss) was used to capture images on a Zeiss upright Axio-Imager fitted with Axio-Cam HRc camera. Images were

pseudocolored to permit overlay, cropped, sized and contrast-brightness enhanced for presentation with ImageJ and Adobe Photoshop software.

## Measurements and cell quantification

Measurements of IHF length were performed using ImageJ v1.51s freeware (National Institutes of Health, Bethesda, USA). The length of the IHF within the interhemispheric midline was determined by comparing Laminin and DAPI-staining. To account for inter-brain variability, this length was then normalized to the entire length of the telencephalon along the interhemispheric midline, which was measured from the caudal-most point of the telencephalon to the rostral edge of cerebral hemispheres.

Cell proliferation and cell death in $Dcc^{kanga}$ MZG was automatically counted using Imaris software (Bitplane) from a region of interest delineated by Glast staining, and excluding the IHF within a single z-slice. Cleaved-caspase 3-positive, TDTOMATO-positive N2A and U251 cells were manually counted using the Cell Counter plugin in ImageJ v1.51s freeware (National Institutes of Health), from a 1187 × 954 μm region of interest.

The number of Sox9-positive cell bodies was counted manually using the Cell Counter plugin in ImageJ v1.51s freeware. Cell proliferation and cell death in tissue were automatically counted using Imaris software (Bitplane) from a region of interest delineated by Glast staining that excluded the IHF in a single z-slice.

The perimeter, circularity and area of U251 and N2A cells were measured from the mean intensity projections of TDTOMATO images following thresholding in ImageJ v1.51s freeware (National Institutes of Health). 48–191 cells per condition were analysed from 3 to 5 biological replicates.

## Fluorescence intensity analysis

To compare fluorescence intensity, tissue sections were processed under identical conditions for immunofluorescence. Fluorescent images at ×20 or ×40 magnification were acquired using identical exposure settings for each fluorescent signal and identical number of slices through the z plane. A multiple intensity projection was created for each z-stack to create a 2D image. Identical regions of interest were outlined in ImageJ freeware, the fluorescence intensity was plotted versus the distance and the average fluorescence intensity was calculated.

## Quantification and statistical analysis

A minimum of three animals were analysed for each separate phenotypic analysis. Sex was not determined for embryonic studies. A mix of male and female adult mice was used to determine the length of the IHF in $Dcc^{kanga}$ and C57Bl/6 mice. All measurements and cell counting were performed on deidentified files so that the researcher remained blind to the experimental conditions. For comparison between two groups, the data was first assessed for normality with a D'Agostino–Pearson omnibus normality test and then statistical differences between two groups were determined either with a parametric Student's t-test or a nonparametric Mann–Whitney test in Prism software (v.6–v.8; GraphPad). To test whether CC or HC length was correlated with IHF length as normalized to total telencephalic midline length, the data was also assessed for normality with D'Agostino–Pearson omnibus normality tests, and a Pearson correlation coefficient was computed if the data was representative of a Gaussian distribution; otherwise, a nonparametric Spearman correlation was performed. For multiple comparisons of cell culture conditions or measurements of GFAP fluorescence across mouse strains, a Kruskal–Wallis test was performed with post-hoc Dunn's multiple comparison test. $p \leq 0.05$ was considered significantly different, where all p values are reported in text. All values are presented as mean ± standard error of the mean (SEM).

## Contact for reagent and resource sharing

Further information and requests for resources and reagents should be directed to and will be fulfilled by Professor Linda J Richards (richards@uq.edu.au).

## Acknowledgements

We thank Marc Tessier-Lavigne for the *Ntn1-lacZ* and *Dcc* knockout mouse lines, and Susan Ackermann (Jackson Laboratory) for the *Dcc^kanga* mouse lines. We thank colleagues for providing the constructs listed. We thank Luke Hammond, Rumelo Amor, Arnaud Guardin, Andrew Thompson and Matisse Jacobs for assistance with microscopy, which was performed in the Queensland Brain Institute's Advanced Microscopy Facility. This work was supported by Australian NHMRC grants GNT456027, GNT631466, GNT1048849 and GNT1126153 to LJR and GNT1059666 to PJL and RJL, and US National Institutes of Health grant 5R01NS058721 to ES and LJR. RS received an Australian Research Council DECRA fellowship (DE160101394). APLM, JWCL, and LM were supported by a research training program scholarship (Australian Postgraduate Award). APLM was further supported by a NHMRC Early Career Research Fellowship (GNT1156820). ALSD was supported by an Australian Postgraduate award. ALSD, JWCL, and LM received a Queensland Brain Institute Top-Up Scholarship. LRF was supported by a UQ Development Fellowship, RJL was supported by a Melbourne Children's Clinician Scientist Fellowship and LJR was supported by an NHMRC Principal Research Fellowship (GNT1120615).

We thank the families and members of the Australian Disorders of the Corpus Callosum (AusDoCC) for their support and time in being involved in this research. We thank the International Research Consortium for the Corpus Callosum and Cerebral Connectivity (IRC5, https://www.irc5.org) researchers for discussions and input.

## Additional information

### Funding

| Funder | Grant reference number | Author |
| --- | --- | --- |
| National Health and Medical Research Council | GNT456027 | Linda J Richards |
| National Health and Medical Research Council | GNT631466 | Linda J Richards |
| National Health and Medical Research Council | GNT1048849 | Linda J Richards |
| National Health and Medical Research Council | GNT1126153 | Linda J Richards |
| National Health and Medical Research Council | GNT1059666 | Richard J Leventer Paul J Lockhart |
| National Institutes of Health | 5R01NS058721 | Elliott H Sherr Linda J Richards |
| Australian Research Council | DE160101394 | Rodrigo Suárez |
| Department of Education, Skills and Employment, Australian Government | Australian Postgraduate Award, Research Training Program | Laura Morcom Ashley PL Marsh Jonathan WC Lim |
| Queensland Brain Institute | Top-Up scholarship | Laura Morcom Jonathan WC Lim Amber-Lee S Donahoo |
| University of Queensland | UQ development fellowship | Laura R Fenlon |
| Murdoch Children's Research Institute | Melbourne Children's Clinician Scientist fellowship | Richard J Leventer |
| National Health and Medical Research Council | GNT1120615 | Linda J Richards |
| National Health and Medical Research Council | GNT1156820 | Ashley PL Marsh |

The funders had no role in study design, data collection and interpretation, or the decision to submit the work for publication.

## Author contributions
Laura Morcom, Conceptualization, Formal analysis, Investigation, Visualization, Methodology, Writing - original draft, Writing - review and editing; Ilan Gobius, Conceptualization, Formal analysis, Supervision, Investigation, Methodology, Writing - review and editing; Ashley PL Marsh, Conceptualization, Resources, Validation, Investigation, Methodology, Writing - original draft, Writing - review and editing; Rodrigo Suárez, Conceptualization, Supervision, Investigation, Writing - review and editing; Jonathan WC Lim, Investigation; Caitlin Bridges, Yunan Ye, Formal analysis, Investigation, Writing - review and editing; Laura R Fenlon, Supervision, Investigation, Writing - review and editing; Yvrick Zagar, Resources, Investigation, Methodology, Writing - review and editing; Amelia M Douglass, Conceptualization, Investigation, Methodology, Writing - review and editing; Amber-Lee S Donahoo, Timothy J Edwards, Investigation, Methodology, Writing - review and editing; Thomas Fothergill, Resources, Methodology, Writing - review and editing; Samreen Shaikh, Investigation, Writing - review and editing; Peter Kozulin, Supervision, Investigation, Methodology, Writing - review and editing; Helen M Cooper, Conceptualization, Resources, Writing - review and editing; IRC5 Consortium, Conceptualization, Validation; Elliott H Sherr, Conceptualization, Formal analysis, Supervision, Funding acquisition, Writing - review and editing; Alain Chédotal, Conceptualization, Resources, Supervision, Funding acquisition, Validation, Writing - review and editing; Richard J Leventer, Paul J Lockhart, Conceptualization, Resources, Supervision, Funding acquisition, Writing - review and editing; Linda J Richards, Conceptualization, Supervision, Funding acquisition, Investigation, Visualization, Methodology, Writing - original draft, Project administration, Writing - review and editing

## Author ORCIDs
Laura Morcom (iD) https://orcid.org/0000-0001-6683-4356
Ashley PL Marsh (iD) http://orcid.org/0000-0001-6049-6931
Rodrigo Suárez (iD) https://orcid.org/0000-0001-5153-5652
Jonathan WC Lim (iD) http://orcid.org/0000-0002-5074-6359
Yunan Ye (iD) https://orcid.org/0000-0001-9084-6314
Laura R Fenlon (iD) https://orcid.org/0000-0002-5132-2419
Amelia M Douglass (iD) https://orcid.org/0000-0001-5398-6473
Peter Kozulin (iD) https://orcid.org/0000-0002-7872-9884
Helen M Cooper (iD) https://orcid.org/0000-0001-9838-5743
Elliott H Sherr (iD) https://orcid.org/0000-0002-4118-5385
Alain Chédotal (iD) https://orcid.org/0000-0001-7577-3794
Paul J Lockhart (iD) http://orcid.org/0000-0003-2531-8413
Linda J Richards (iD) https://orcid.org/0000-0002-7590-7390

## Ethics
Human subjects: Ethics for human experimentation was acquired by local ethics committees at The University of Queensland (Australia), the Royal Children's hospital (Australia), and UCSF Benioff Children's Hospital (USA). The research was carried out in accordance with the provisions contained in the National Statement on Ethical Conduct in Human Research (USA) under IRB number 10-01008 and with the regulations governing experimentation on humans (Australia), under the following human ethics approvals: HEU 2007/163 (previously 2006000899), HEU 2014000535, HEU 2015001306.

Animal experimentation: Prior approval for all breeding and experiments was obtained from the University of Queensland Animal Ethics Committee and was conducted in accordance with the Australian code for the care and use of animals for scientific purposes. The protocol, experiments and animal numbers were approved under the following project approval numbers: (QBI/305/17, QBI/311/14 NHMRC (NF), QBI/356/17, QBI/306/17, and QBI/240/14/MDF (NF)).

## Decision letter and Author response
Decision letter https://doi.org/10.7554/eLife.61769.sa1
Author response https://doi.org/10.7554/eLife.61769.sa2

## Additional files

### Supplementary files

• Supplementary file 1. Statistics related to quantified data in *Figures 1–8* and *Figure 1—figure supplement 1–Figure 7—figure supplement 1*. CA: cell area; CP: cell perimeter; CC3: cleaved caspase 3; DCCK: DCCKanga; E: embryonic day; EP: electroporated; exp: experimental; FI: fluorescence intensity; IGG: indusium griseum glia; MZG: midline zipper glia; p: postnatal day; ROI: region of interest; TDT: TDTOMATO; vs.: versus; wt: wildtype.

• Transparent reporting form

### Data availability

All data generated or analysed during this study are included in the manuscript and supporting files. Source data files have been provided for all figures that contain numerical data.

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

# Appendix 1

**Appendix 1—key resources table**

| Reagent type (species) or resource | Designation | Source or reference | Identifiers | Additional information |
|---|---|---|---|---|
| Gene (*Homo sapiens*) | *DCC* | NCBI | 1630 | |
| Gene (*Mus musculus*) | *Dcc* | NCBI | 13176 | |
| Gene (*Mus musculus*) | *Ntn1* | NCBI | 18208 | |
| Strain, strain background (*Mus musculus*) | *Dcc*$^{flox/flox}$, C57BL/6J | *Krimpenfort et al., 2012* | N/A | |
| Strain, strain background (*Mus musculus*) | *Dcc*$^{kanga}$, C57BL/6J | *Finger et al., 2002* | N/A | |
| Strain, strain background (*Mus musculus*) | *Dcc*$^{-/-}$, C57BL/6J | *Fazeli et al., 1997* | N/A | |
| Strain, strain background (*Mus musculus*) | *Emx1*$^{iCre}$, C57BL/6J | *Kessaris et al., 2006* | N/A | |
| Strain, strain background (*Mus musculus*) | *Ntn1-lacZ*, C57BL/6J | *Serafini et al., 1996* | N/A | |
| Strain, strain background (*Mus musculus*) | *tdTomato*$^{flox\_stop}$, C57BL/6J | *Madisen et al., 2010* | N/A | |
| Cell line (*Homo sapiens*) | HEK293T | ATCC | RRID:CVCL_0045 | ATCC Cat# CRL-1573 |
| Cell line (*Homo sapiens*) | U251MG | ATCC | RRID:CVCL_0021 | Obtained as U-373MG (RRID:CVCL_2219) but subsequently identified as U-251 via PCR-based short tandem repeat profiling |
| Cell line (*Mus musculus*) | Neuro-2A (N2A) | ATCC | RRID:CVCL_0470 | Obtained via the University of Queensland |
| Cell line (*Chlorocebus aethiops*) | COS-7 | ATCC | RRID:CVCL_0224 | ATCC Cat# CRL-1651 |
| Antibody | Goat polyclonal anti-DCC | Santa Cruz Biotechnology | sc-6535, RRID:AB_2245770 | (1:200) western blot; (1:500) immunofluorescence |
| Antibody | Goat polyclonal anti-NTN1 | R&D Systems | AF1109, RRID:AB_2298775 | (1:500) western blot; (1:500) immunofluorescence |
| Antibody | Rabbit monoclonal anti-GADPH | Cell Signaling Technology | 2118, RRID:AB_561053 | (1:2000) western blot |
| Antibody | Rabbit polyclonal anti-GADPH | IMGENEX | IMG-5143A, RRID:AB_613387 | (1:1000) western blot |
| Antibody | Rabbit polyclonal anti-APC | Abcam | ab15270, RRID:AB_301806 | (1:250) |

*Continued on next page*

*Appendix 1—key resources table continued*

| Reagent type (species) or resource | Designation | Source or reference | Identifiers | Additional information |
|---|---|---|---|---|
| Antibody | Mouse monoclonal anti-α-DAG1 | Merck | 05-593, RRID:AB_309828 | (1:250) |
| Antibody | Rabbit polyclonal anti-β-catenin | Cell Signaling Technology | 9562, RRID:AB_331149 | (1:500) |
| Antibody | Mouse monoclonal anti-β-dystroglycan (MANDAG2) | Developmental Studies Hybridoma Bank | 7D11, RRID:AB_2211772 | (1:50) |
| Antibody | Chicken polyclonal anti-β-galactosidase | Abcam | ab9361, RRID:AB_307210 | (1:500) |
| Antibody | Rabbit polyclonal anti-cleaved-caspase 3 | Cell Signaling Technology | 9661, RRID:AB_2341188 | (1:500) |
| Antibody | Goat polyclonal anti-DCC | Santa Cruz Biotechnology | sc-6535, RRID:AB_2245770 | (1:500) |
| Antibody | Mouse monoclonal anti-GAP43 | Millipore | MAB347, RRID:AB_94881 | (1:500) |
| Antibody | Mouse monoclonal anti-GFAP | Millipore | MAB3402, RRID:AB_94844 | (1:500) |
| Antibody | Rabbit polyclonal anti-GFAP | Dako | Z0334, RRID:AB_10013382 | (1:500) |
| Antibody | Mouse monoclonal anti-Glast (EAAT1) | Abcam | Ab49643, RRID:AB_869830 | (1:500) |
| Antibody | Rabbit polyclonal anti-Glast (EAAT1) | Abcam | Ab416, RRID:AB_304334 | (1:250) |
| Antibody | Mouse monoclonal anti-KI67 | BD Pharmingen | 550609, RRID:AB_393778 | (1:500) |
| Antibody | Chicken polyclonal anti-Laminin | LS-Bio | C96142, RRID:AB_2033342 | (1:500) |
| Antibody | Rabbit polyclonal anti-Laminin (pan-Laminin) | Sigma | L9393, RRID:AB_477163 | (1:500) |
| Antibody | Mouse monoclonal anti-N-cadherin (CDH2) | BD Biosciences | 610921, RRID:AB_398236 | (1:250) |
| Antibody | Rat monoclonal anti-Nestin (NES) | Developmental Studies Hybridoma Bank | AB 2235915, RRID:AB_2235915 | (1:50) |
| Antibody | Chicken polyclonal anti-Nestin (NES) | Abcam | Ab134017, RRID:AB_2753197 | (1:1000) |
| Antibody | Goat polyclonal anti-NTN1 | R&D Systems | AF1109, RRID:AB_2298775 | (1:500) |

*Continued on next page*

*Appendix 1—key resources table continued*

| Reagent type (species) or resource | Designation | Source or reference | Identifiers | Additional information |
|---|---|---|---|---|
| Antibody | Mouse monoclonal anti-neurofilament | Millipore | MAB1621, RRID:AB_94294 | (1:500) |
| Antibody | Rabbit polyclonal anti-NFIA | Aviva Systems Biology | ARP32714, RRID:AB_576739 | (1:500) |
| Antibody | Rabbit polyclonal anti-NFIB | Sigma | HPA003956, RRID:AB_1854424 | (1:500) |
| Antibody | Rabbit polyclonal anti-neuronal-specific-ßIII-tubulin (TUBB3) | Abcam | Ab18207, RRID:AB_444319 | (1:500) |
| Antibody | Rabbit polyclonal anti-phospho p44/42 MAPK (ERK1/2) | Cell Signaling Technology | 9101, RRID:AB_331646 | (1:250) |
| Antibody | Rabbit polyclonal anti-SOX9 | Merck | AB5535, RRID:AB_2239761 | (1:500) |
| Antibody | Goat polyclonal anti-TDTOMATO | Sicgen | Ab8181-200, RRID:AB_2722750 | (1:500) |
| Recombinant DNA reagent | pCAG-TDTOMATO | This paper | | |
| Recombinant DNA reagent | pCAG-DCC:TDTOMATO | This paper | | |
| Recombinant DNA reagent | pCAG-DCC$^{KANGA}$:TDTOMATO | This paper | | |
| Recombinant DNA reagent | pCAG-DCC$^{M743L}$:TDTOMATO | This paper | | |
| Recombinant DNA reagent | pCAG-DCC$^{V754M}$:TDTOMATO | This paper | | |
| Recombinant DNA reagent | pCAG-DCC$^{A893T}$:TDTOMATO | This paper | | |
| Recombinant DNA reagent | pCAG-DCC$^{V793G}$:TDTOMATO | This paper | | |
| Recombinant DNA reagent | pCAG-DCC$^{MG805E}$:TDTOMATO | This paper | | |
| Recombinant DNA reagent | pCAG-DCC$^{M1217V;A1250T}$:TDTOMATO | This paper | | |
| Recombinant DNA reagent | pCAG-H2B-GFP-2A-MyrTDTOMATO | Arnold Kriegstein (UCSF) | | |
| Recombinant DNA reagent | p-SUPER-*Dcc*-shRNA | Xiong Zhiqi; *Zhang et al., 2018* | | |
| Recombinant DNA reagent | pCAG-*Dcc*-CRISPR 1 | Atum; this paper | | Targeting chr18:71954969–71955009 |
| Recombinant DNA reagent | pCAG-*Dcc*-CRISPR 2 | Atum; this paper | | Targeting chr18:71826146–71826092 |
| Recombinant DNA reagent | *Fgf8* cDNA | Gail Martin, UCSF | | In situ hybridization riboprobe |
| Recombinant DNA reagent | *Ntn1* cDNA | Helen Cooper | | In situ hybridization riboprobe |

*Continued on next page*

*Appendix 1—key resources table continued*

| Reagent type (species) or resource | Designation | Source or reference | Identifiers | Additional information |
|---|---|---|---|---|
| Recombinant DNA reagent | *Mmp2* cDNA | This paper | | In situ hybridization riboprobe |
| Sequence-based reagent | *Mmp-2* cDNA forward primer | Allen Brain Atlas | | 5'-ATGGTGACCAA GAACAGAAGGT |
| Sequence-based reagent | *Mmp-2* cDNA reverse primer | Allen Brain Atlas | | 5'-AATCACTGCTA CAATCACCACG |
| Sequence-based reagent | *Dcc* site-directed mutagenesis p.Met743Leu forward primer | This paper | | 5'-GAGGAGGTGTCC AACTCAAGATGAT ACAGTTTGTCTG |
| Sequence-based reagent | *Dcc* site-directed mutagenesis p.Met743Leu reverse primer | This paper | | 5'-CAGACAAACTG TATCATCTTGAG TTGGACACCTCCTC |
| Sequence-based reagent | *Dcc* site-directed mutagenesis p.Val754Met forward primer | This paper | | 5'-TAATATAGCCTCTC ACCATGATGTTT GGGTTGAGAGG |
| Sequence-based reagent | *Dcc* site-directed mutagenesis p.Val754Met reverse primer | This paper | | 5'-CCTCTCAACCCAAAC ATCATGGTGAGAG GCTATATTA |
| Sequence-based reagent | *Dcc* site-directed mutagenesis p.Ala893Thr forward primer | This paper | | 5'- ACTTGTACTTGGT ACTGGCAGAAA AGCTGGTCCT |
| Sequence-based reagent | *Dcc* site-directed mutagenesis p.Ala893Thr reverse primer | This paper | | 5'-AGGACCAGCTTTTC TGCCAGTACC AAGTACAAGT |
| Sequence-based reagent | *Dcc* site-directed mutagenesis p.Val793Gl forward primer | This paper | | 5'-ACTAGAGTCGAGTTCT CATTATGGAATCTC CTTAAAAGCTTTCAAC |
| Sequence-based reagent | *Dcc* site-directed mutagenesis p.Val793Gl reverse primer | This paper | | 5'-GTTGAAAGCTTTTAAG GAGATTCCATAATG AGAACTCGACTCTAGT |
| Sequence-based reagent | *Dcc* site-directed mutagenesis p.Gly805Glu forward primer | This paper | | 5'-CACTTTCGTAGA GAGGGACCTCTTC TCCGGCATTGTTGAA |
| Sequence-based reagent | *Dcc* site-directed mutagenesis p.Gly805Glu reverse primer | This paper | | 5'-TTCAACAATGCCG GAGAAGAGGTCC CTCTCTACGAAAGTG |
| Sequence-based reagent | *Dcc* site-directed mutagenesis p.Met1217Val;p.Ala1250Thr forward 1 primer | This paper | | 5'-GTTCCAAAGTG GACACGGAGCTG CCTGCGTC |
| Sequence-based reagent | *Dcc* site-directed mutagenesis p.Met1217Val;p.Ala1250Thr reverse 1 primer | This paper | | 5'-GACGCAGGCAG CTCCGTGTCCAC TTTGGAAC |
| Sequence-based reagent | *Dcc* site-directed mutagenesis p.Met1217Val;p.Ala1250Thr forward 2 primer | This paper | | 5'-GTACAGGGAT GGTACTCACAA CAGCAGGATTACTGG |
| Sequence-based reagent | *Dcc* site-directed mutagenesis p.Met1217Val;p.Ala1250Thr reverse 2 primer | This paper | | 5'-CCAGTAATCCT GCTGTTGTGAGTA CCATCCCTGTAC |
| Sequence-based reagent | *Dcc* site-directed mutagenesis p.Val848Arg forward primer | This paper | | 5'-CAGCCTGTACAC CTCTTGGTGGGA GCATGGGGG |
| Sequence-based reagent | *Dcc* site-directed mutagenesis p.Val848Arg reverse primer | This paper | | 5'-CCCCCATGCTC CCACCAAGAGGT GTACAGGCTG |

*Continued on next page*

*Appendix 1—key resources table continued*

| Reagent type (species) or resource | Designation | Source or reference | Identifiers | Additional information |
|---|---|---|---|---|
| Sequence-based reagent | *Dcc* site-directed mutagenesis p.His857Ala forward primer | This paper | | 5'-ACCCTCACAGCCTCAG CGGTAAGAGCCACAGC |
| Sequence-based reagent | *Dcc* site-directed mutagenesis p.His857Ala reverse primer | This paper | | 5'-GCTGTGGCTCTTACC GCTGAGGCTGTGAGGG |
| Sequence-based reagent | *Dcc* site-directed mutagenesis p.p.del-P3 (Kanga) forward primer | This paper | | 5'-CCACAGAGGATCC AGCCAGTGGAGATCCACC |
| Sequence-based reagent | *Dcc* site-directed mutagenesis p.p.del-P3 (Kanga) reverse primer | This paper | | 5'-GGTGGATCTCCA CTGGCTGGATCCTCTGTGG |
| Peptide, recombinant protein | *Ntn1* | R&D Systems | 1109-N1 | 100 ng/mL |
| Peptide, recombinant protein | *NTN1-AP* | This paper | | Generated as supernatant from HEK293T as previously described in *Zelina et al., 2014* |
| Commercial assay, kit | Click-iT EdU Alexa Fluor 488 Imaging Kit | Invitrogen | C10337 | |
| Commercial assay, kit | QuickChange II Site-Directed Mutagenesis Kit | Stratagene | 200524 | |
| Software, algorithm | Fiji | Fiji | RRID:SCR_ 002285 | |
| Software, algorithm | Prism | GraphPad | RRID:SCR_ 002798 | |
| Software, algorithm | Imaris | Bitplane | RRID:SCR_ 007370 | |

