## [Decision Letter]

**Acceptance summary:**

Your study is a welcome follow-up to your previous demonstration that midline zipper glia (MZG) migrate along the interhemispheric fissure (IHF) and intercalate across the hemispheres, and in doing so, remodel the meningeal basement membrane to provide a substrate for callosal axon growth. You identify DCC and its ligand Netrin1 to be important for this process, by acting on the distribution and morphology of MZG. Importantly, human subjects with DCC mutations display disrupted IHF remodelling associated with corpus callosum and hippocampal commissure malformations. Your work thus nicely establishes that, in addition to service as axon guidance signals for callosal axons attraction to and across the midlin, DCC and Netrin-1 are crucial in callosal formation and interhemispheric fusion.

**Decision letter after peer review:**

Thank you for submitting your article "DCC regulates astroglial development essential for telencephalic morphogenesis and corpus callosum formation" for consideration by *eLife*. Your article has been reviewed by 3 peer reviewers, and the evaluation has been overseen by a Reviewing Editor and Huda Zoghbi as the Senior Editor. The following individuals involved in review of your submission have agreed to reveal their identity: Samantha J Butler (Reviewer #2); Thomas Pratt (Reviewer #3).

The reviewers have discussed the reviews with one another and the Reviewing Editor has drafted this decision to help you prepare a revised submission.

Summary:

This study is a welcome follow-up to your earlier demonstration that midline zipper glia (MZG) migrate along the interhemispheric fissure (IHF) and intercalate across the hemispheres, and in doing so, remodel the meningeal basement membrane to provide a substrate for callosal axon growth. You identify DCC and its ligand Netrin1 to be important for this process, by acting on the distribution and morphology of MZG, in addition to their service as axon guidance signals for callosal axons to be attracted to and across the midline.

The reviewers were very favorable about the manuscript in the reviews and in the off-line Consultation Session. Reviewers 1 and 2 asked for several modifications of your data presentation, but do not ask for new data (except for the first point below, optional). Below are some key points to address, and others that you can see in the appended reviews.

Essential revisions:

1. Reviewer 2 thought that you should consider removing the shRNA approach from the manuscript and re-focusing the cKO data on a description of a Dcc phenotypic series, to better fit with the initial description of lack of interhemispheric remodeling observed in Dcc/Ntn mutant mice, and relate this to phenotypes observed in patients. Reviewer 1 thought you could simply downplay some of the focus on cell autonomy since the shRNA/CRISPR and conditional KO did not give a particularly clear answer. It is your preference as to which to do.

2. Reviewer 1 suggested that to lend more credence to a suspected causal relationship between the severity of callosal agenesis and the extent to which the IHF had been remodeled, it would be informative to generate scatterplots of IHF length vs. CC/HC length.

3. On the large increase in the total number of Sox9+ cells along the IHF by E16: Reviewer 1 asks for an possible explanation, while Reviewer 2 wonders whether the increase might be due to an absence of a stop signal at the midline.

4. Both Reviewers 1 and 2 ask why simply transfecting WT DCC into cell lines results in such a dramatic change in morphology but addition of NTN1 does not. Have you checked whether the cell lines express Netrin1?

*Reviewer #1:*

In this manuscript, the authors revisit DCC and NTN1 mutants in order to better define the basis for midline crossing defects. This group recent demonstrated that midline zipper glia (MZG) must migrate along the interhemispheric fissure (IHF) and intercalate across the midline while remodeling the meningeal basement membrane to provide a substrate for callosal axons to cross the midline. In this study, they show that DCC and its ligand NTN1 are required for proper midline zipper glia (MZG) distribution/morphology along the IHF, proper remodeling of the basement membrane, and subsequent corpus callosum (CC) formation. The data in figures 2 and 3 generally do a nice job of supporting the model that DCC and NTN1 are expressed in MZG and that the morphology and distribution of MZG are affected in DCC/NTN1 mutants. There appear to be some defects in MZG migration that may account for this (Figure 4). Due to technical limitations, the author's attempt to use a conditional knockout of DCC to genetically dissect whether CC formation defects are due to defects in MZG or callosal axons are a bit inconclusive (Figure 6). Finally, the paper ends with experiments showing that mutations in DCC identified in acallosal patients are loss-of-function using an in vitro cell morphology assay (Figure 7 and 8).

The authors are commended for the quality of their imaging data and for being as quantitative as possible when measuring their in vivo phenotypes, which is not often done with these types of studies. There are few issues that need to be addressed.

1. In Figure 4, in addition to the migration defects of Sox9+ MZG, there seems to be a rather large increase in the total number of Sox9+ cells along the IHF by E16 (more than 2 fold, Figure 4G). The authors show there is no change in cell cycle or apoptosis of these cells in the supplemental data (Figure S4), so what accounts for this increase? Is this also seen with NFIA/B staining at E16?

2. Regarding the attempt to distinguish between DCC in MZG versus callosal axons (Figure 6), the incomplete deletion/loss of DCC protein (Figures 6C, I, J) is a bit concerning. It’s not clear to me why this would happen, but it confounds the interpretation of the results. While the authors state "The severity of callosal agenesis was associated with the extent to which the IHF had been remodeled" (pg 15), they don't actually quantify this. It might be informative to generate scatterplots of IHF length vs. CC/HC length to determine if there is a significant correlation between the two. This might lend more evidence to a causal relationship between IHF remodeling and CC/HC formation.

3. At the end of the result section, the authors state: "mutations that affect the ability for DCC to regulate cell shape (Figure 8F), are likely to cause callosal agenesis through perturbed MZG migration and IHF remodelling." (pg. 19). While the authors nicely show that patient mutations in DCC affect the morphology of cell in cell lines (Figure 7-8), it is not clear why simply transfecting WT DCC into cell lines results in such a dramatic change in morphology, or why addition of NTN1 doesn't increase this. The authors mention that the cell lines could express NTN1 or that NTN1 is not required for the effect. This seems an important distinction. Did the authors check this? Could they use a function blocking antibody or a soluble fragment of the NTN1 binding domain of DCC to block NTN1:DCC interactions? DCC has been shown to function as a "dependence receptor" that can induce apoptosis in the absence of ligand, are the authors certain that the morphology changes they are seeing in DCC transfected cells aren't cytoskeletal changes resulting from caspase activation?

*Reviewer #2:*

This paper is the second in a series of landmark studies from the Richards lab that re-assess the molecular and cellular mechanisms that permit the corpus callosum (CC) to cross the interhemispheric midline in the telencephalon. The Richards lab previously showed key role for a specialized population of fetal astrocytes, the midline zipper glia (MZG), establishing this substrate when the MZG migrate into the interhemispheric fissure (IHF), intercalate with one another and degrade the intervening leptomeninges. In this manuscript, the authors now assess the requirement for the Ntn1/Dcc pathway in remodeling the IHF. In an elegant series of experiments, they shown that Ntn1/Dcc regulate the migration pathway of the MZG, potentially by directly controlling cytoskeletal dynamics. This mechanism is conserved between humans and rodents; the authors show that Dcc mutations that cause CC dysgenesis in humans, cause striking changes in the morphology of astroglial-like cells, consistent with the regulation of MZG migration. Thus, Dcc appears to have two roles first, remodeling the MGZ and then guiding CC axons towards the telencephalic midline. Together, these studies continue the overgoing re-evaluation of the role of netrin1/Dcc in establishing neural circuitry, and shed further understanding on a fascinating and beautiful piece of biology.

This is a very beautiful manuscript, the authors are to be congratulated for the very high quality of their images, and detailed quantifications. Would that all studies were so thorough. These studies will be of great interest to the developmental neuroscience research and clinical communities. I strongly support publication.

The authors should be congratulated by including what was clearly a difficult conditional analysis to assess whether Dcc is required in the callosal axons, or in the MZG radial fibers. This analysis was confounded (a) by the low efficiency of the shRNA to knock down Dcc and (b) the mosaic nature of Emx::cre line, which appears to be variably expressing cre in both callosal neurons and MZG, given that TDT/Dcc are present in both axons (Figure 5B), and the MZG (Figure 5O) in the less severely affected animals.

As currently presented, however, the analysis (sadly) does not greatly add to the paper, since technical issues beyond the authors' control, have made it difficult to assess specifically where Dcc is required with much confidence. Would the authors could consider removing the shRNA approach from the manuscript, and re-focusing the cKO data on a description of a Dcc phenotypic series? This analysis might fit better with the initial description of lack of interhemispheric remodeling observed in Dcc/Ntn mutant mice, and how they relates to (variable?) phenotypes observed in patients.

*Reviewer #3:*

This is a very thorough study giving new insight into a non-cell autonomous mechanism for DCC in axon guidance in midline fusion important for corpus callosum axon guidance.

I have no substantive concerns.

---

## [Author Response]

Essential revisions:1. Reviewer 2 thought that you should consider removing the shRNA approach from the manuscript and re-focusing the cKO data on a description of a Dcc phenotypic series, to better fit with the initial description of lack of interhemispheric remodeling observed in Dcc/Ntn mutant mice, and relate this to phenotypes observed in patients. Reviewer 1 thought you could simply downplay some of the focus on cell autonomy since the shRNA/CRISPR and conditional KO did not give a particularly clear answer. It is your preference as to which to do.

We thank the reviewers for raising this point and for understanding the experimental limitations inherent in this system. The Results section entitled, ‘Variable DCC knockdown during midline development causes a spectrum of callosal phenotypes’, has been revised (page 14-18) to reduce the focus on cell autonomy. The text describing the shRNA approaches has been reduced considerably, shifting the focus to the cKO data as both reviewers suggested. We also include scatterplots showing a strong relationship between IHF remodelling and CC/HC length as reviewer 1 suggested (in point 2).

2. Reviewer 1 suggested that to lend more credence to a suspected causal relationship between the severity of callosal agenesis and the extent to which the IHF had been remodeled, it would be informative to generate scatterplots of IHF length vs. CC/HC length.

We thank the reviewer for this suggestion and have included this analysis in the manuscript. Scatterplots have been included in figure 6 and figure S6 and statistics have been described in the text on page 15 and updated in supplementary table 1. We report a significant correlation between IHF length and CC or HC length. References to panels in figure 6 were amended throughout the results on page 15-17 to include these changes.

3. On the large increase in the total number of Sox9+ cells along the IHF by E16: Reviewer 1 asks for an possible explanation, while Reviewer 2 wonders whether the increase might be due to an absence of a stop signal at the midline.

We appreciate the reviewers’ interest in this striking phenotype. Our quantification of cell proliferation, cell death and total differentiated GFAP-positive MZG suggest that a similar number of MZG cells are generated and differentiate between our mouse models. However, although Sox9+ cell migration is delayed at earlier ages, it does increase at E16 suggesting a delay in the migration of these cells. Once they arrive at the midline, they are unable to undertake IHF remodelling due to further defects in differentiation mediated by DCC/Netrin 1 signalling. Thus, more MZG are present at the base of the IHF at E16 than wildtype, which migrate rostral and dorsal for further IHF remodelling. We have amended the text on page 12 to further clarify this point as such:

“*Dcc^kanga^* MZG remain adjacent to the unremodelled IHF at E16 and do not undergo further maturation to enable IHF remodelling. In contrast, wildtype MZG at E16 are scattered along the midline where IHF remodelling has already occurred and continue to expand their domain rostral and dorsal for further IHF remodelling (Gobius et al., 2016).”

4. Both Reviewers 1 and 2 ask why simply transfecting WT DCC into cell lines results in such a dramatic change in morphology but addition of NTN1 does not. Have you checked whether the cell lines express Netrin1?

We thank the reviewers for highlighting this. In response to the reviewer’s suggestion, we performed western blots of NTN1 expression. These are shown in Figure 7—figure supplement 1j and demonstrate that these cell lines indeed endogenously express NTN1. We include this new result on page 18.

“Interestingly, application of NTN1 did not affect cell shape following DCC expression in either cell line (Supplementary File 1 and Figure 7A-F), suggesting that endogenous NTN1, which is known to be expressed by U251 cells (Chen et al., 2017), may be sufficient for activation of DCC:TDTOMATO receptors, or that NTN1 is not required for this effect. […] Thus, addition of DCC induced cytoskeletal rearrangements in both N2A and U251 cells, which may involve autocrine NTN1 signalling.”

We have also updated the methods, raw data file and Supplementary file 1 to reflect these changes.

Reviewer #1:In this manuscript, the authors revisit DCC and NTN1 mutants in order to better define the basis for midline crossing defects. This group recent demonstrated that midline zipper glia (MZG) must migrate along the interhemispheric fissure (IHF) and intercalate across the midline while remodeling the meningeal basement membrane to provide a substrate for callosal axons to cross the midline. In this study, they show that DCC and its ligand NTN1 are required for proper midline zipper glia (MZG) distribution/morphology along the IHF, proper remodeling of the basement membrane, and subsequent corpus callosum (CC) formation. The data in figures 2 and 3 generally do a nice job of supporting the model that DCC and NTN1 are expressed in MZG and that the morphology and distribution of MZG are affected in DCC/NTN1 mutants. There appear to be some defects in MZG migration that may account for this (Figure 4). Due to technical limitations, the author's attempt to use a conditional knockout of DCC to genetically dissect whether CC formation defects are due to defects in MZG or callosal axons are a bit inconclusive (Figure 6). Finally, the paper ends with experiments showing that mutations in DCC identified in acallosal patients are loss-of-function using an in vitro cell morphology assay (Figure 7 and 8).The authors are commended for the quality of their imaging data and for being as quantitative as possible when measuring their in vivo phenotypes, which is not often done with these types of studies. There are few issues that need to be addressed.1. In Figure 4, in addition to the migration defects of Sox9+ MZG, there seems to be a rather large increase in the total number of Sox9+ cells along the IHF by E16 (more than 2 fold, Figure 4G). The authors show there is no change in cell cycle or apoptosis of these cells in the supplemental data (Figure S4), so what accounts for this increase? Is this also seen with NFIA/B staining at E16?

Please refer to the “Essential revisions” comment number 3 (above). NFIA and NFIB immunostaining are less-specific for MZG at later ages, since they are also expressed by neurons and oligodendrocyte precursors, therefore we have not quantified NFIA/NFIB positive cells at E16 since they do not exclusively represent the SOX9-positive population.

2. Regarding the attempt to distinguish between DCC in MZG versus callosal axons (Figure 6), the incomplete deletion/loss of DCC protein (Figures 6C, I, J) is a bit concerning. It’s not clear to me why this would happen, but it confounds the interpretation of the results. While the authors state "The severity of callosal agenesis was associated with the extent to which the IHF had been remodeled" (pg 15), they don't actually quantify this. It might be informative to generate scatterplots of IHF length vs. CC/HC length to determine if there is a significant correlation between the two. This might lend more evidence to a causal relationship between IHF remodeling and CC/HC formation.

Please see “Essential revisions number 2” for our response.

3. At the end of the result section, the authors state: "mutations that affect the ability for DCC to regulate cell shape (Figure 8F), are likely to cause callosal agenesis through perturbed MZG migration and IHF remodelling." (pg. 19). While the authors nicely show that patient mutations in DCC affect the morphology of cell in cell lines (Figure 7-8), it is not clear why simply transfecting WT DCC into cell lines results in such a dramatic change in morphology, or why addition of NTN1 doesn't increase this. The authors mention that the cell lines could express NTN1 or that NTN1 is not required for the effect. This seems an important distinction. Did the authors check this? Could they use a function blocking antibody or a soluble fragment of the NTN1 binding domain of DCC to block NTN1:DCC interactions? DCC has been shown to function as a "dependence receptor" that can induce apoptosis in the absence of ligand, are the authors certain that the morphology changes they are seeing in DCC transfected cells aren't cytoskeletal changes resulting from caspase activation?

Please see Essential revisions number 4 for our response regarding NTN1 expression in the cell lines. In addition, we performed immunohistochemistry for cleaved-caspase3, a marker of apoptosis but found very little apoptosis occurring in our cultures, and no difference in expression was observed between groups. We have included this new data in Supplementary figure 7I, addressed this in the text on page 18, and updated the methods (page 28), supplementary table 1 and raw data file to reflect this addition.

Reviewer #2:This paper is the second in a series of landmark studies from the Richards lab that re-assess the molecular and cellular mechanisms that permit the corpus callosum (CC) to cross the interhemispheric midline in the telencephalon. The Richards lab previously showed key role for a specialized population of fetal astrocytes, the midline zipper glia (MZG), establishing this substrate when the MZG migrate into the interhemispheric fissure (IHF), intercalate with one another and degrade the intervening leptomeninges. In this manuscript, the authors now assess the requirement for the Ntn1/Dcc pathway in remodeling the IHF. In an elegant series of experiments, they shown that Ntn1/Dcc regulate the migration pathway of the MZG, potentially by directly controlling cytoskeletal dynamics. This mechanism is conserved between humans and rodents; the authors show that Dcc mutations that cause CC dysgenesis in humans, cause striking changes in the morphology of astroglial-like cells, consistent with the regulation of MZG migration. Thus, Dcc appears to have two roles first, remodeling the MGZ and then guiding CC axons towards the telencephalic midline. Together, these studies continue the overgoing re-evaluation of the role of netrin1/Dcc in establishing neural circuitry, and shed further understanding on a fascinating and beautiful piece of biology.This is a very beautiful manuscript, the authors are to be congratulated for the very high quality of their images, and detailed quantifications. Would that all studies were so thorough! These studies will be of great interest to the developmental neuroscience research and clinical communities. I strongly support publication.The authors should be congratulated by including what was clearly a difficult conditional analysis to assess whether Dcc is required in the callosal axons, or in the MZG radial fibers. This analysis was confounded (a) by the low efficiency of the shRNA to knock down Dcc and (b) the mosaic nature of Emx::cre line, which appears to be variably expressing cre in both callosal neurons and MZG, given that TDT/Dcc are present in both axons (Figure 5B), and the MZG (Figure 5O) in the less severely affected animals.As currently presented, however, the analysis (sadly) does not greatly add to the paper, since technical issues beyond the authors' control, have made it difficult to assess specifically where Dcc is required with much confidence. Would the authors could consider removing the shRNA approach from the manuscript, and re-focusing the cKO data on a description of a Dcc phenotypic series? This analysis might fit better with the initial description of lack of interhemispheric remodeling observed in Dcc/Ntn mutant mice, and how they relates to (variable?) phenotypes observed in patients.

Please see Essential revisions number 1 for our full response. The Results section entitled, ‘Variable DCC knockdown during midline development causes a spectrum of callosal phenotypes’, has been revised (page 14-18) to reduce the focus on cell autonomy. The text describing the shRNA approaches has been reduced considerably, shifting the focus to the cKO data as both reviewers suggested.